# CREDIT: CERTIFIED OWNERSHIP VERIFICATION OF DEEP NEURAL NETWORKS AGAINST MODEL EXTRACTION ATTACKS

## ABSTRACT

Machine Learning as a Service (MLaaS) has emerged as a widely adopted paradigm for providing access to deep neural network (DNN) models, enabling users to conveniently leverage these models through standardized APIs. However, such services are highly vulnerable to Model Extraction Attacks (MEAs), where an adversary repeatedly queries a target model to collect input–output pairs and uses them to train a surrogate model that closely replicates its functionality. While numerous defense strategies have been proposed, verifying the ownership of a suspicious model with strict theoretical guarantees remains a challenging task. To address this gap, we introduce CREDIT, a certified ownership verification against MEAs. Specifically, we employ mutual information to quantify the similarity between DNN models, propose a practical verification threshold, and provide rigorous theoretical guarantees for ownership verification based on this threshold. We extensively evaluate our approach on several mainstream datasets across different domains and tasks, achieving state-of-the-art performance. Our implementation is publicly available at: `https://anonymous.4open.science/r/CREDIT`.

## 1 INTRODUCTION

Machine Learning as a Service (MLaaS) (Ribeiro et al., 2015; Weng et al., 2022) is a modern model deployment solution that has been widely adopted for various state-of-the-art models, such as Google Cloud Vision in computer vision (Mulfari et al., 2016; Bisong, 2019), OpenAI ChatGPT in large language models (Achiam et al., 2023; Hurst et al., 2024), and financial forecasting in time-series analysis (Sezer et al., 2020; Cheng et al., 2022). The development of deep neural networks (DNNs) demands carefully designed architectures, proprietary datasets, extensive computational resources, and substantial training time, rendering the resulting models both expensive to build and critical intellectual property (Aristodemou & Tietze, 2018; Lederer et al., 2023; Cottier et al., 2024). To safeguard these properties while ensuring scalable accessibility, model owners increasingly adopt MLaaS paradigm, where models are hosted on cloud platforms and accessed through APIs. This deployment strategy allows owners to safeguard their models from direct distribution while leveraging the computational advantages of cloud infrastructure, and at the same time provides users with convenient access to state-of-the-art models (Cusumano, 2010; Gibson et al., 2012). However, the MLaaS deployment paradigm carries significant potential risks, one of the most serious of which is the Model Extraction Attacks (MEAs) (Tramèr et al., 2016; Wang & Gong, 2018). In an MEA, an adversary queries the target model through its API to obtain input–output pairs, then uses these pairs to train a surrogate model that closely replicates the target model's functionality. This enables the attacker to acquire advanced models at only minimal cost compared with training the original model from scratch (Wang et al., 2022; Zhao et al., 2025; Orekondy et al., 2019a). Numerous studies have demonstrated that MLaaS is highly vulnerable to MEAs, posing a severe threat to model owners, whose valuable intellectual property faces a substantial risk of theft (He et al., 2021; Li et al., 2024). Therefore, there is an urgent need for effective defense mechanisms against MEAs to ensure the security of model owners' intellectual property (Xue et al., 2021; Lederer et al., 2023).

Numerous studies have been proposed to defend against MEAs. Current ownership verification based defense approaches can be broadly categorized into two techniques: watermarking (Tan et al.,

2023; Boenisch, 2021; Adi et al., 2018; Jia et al., 2021) and fingerprinting (Cao et al., 2021; Peng et al., 2022; Guan et al., 2022; Waheed et al., 2024). Watermarking techniques refer to embedding specially designed input-output pairs into a model. If the model owner can successfully verify these pairs, ownership of the model can be asserted. However, watermarking-based methods often cause significant performance compromises because they introduce task-irrelevant information, which can degrade performance on the downstream tasks (Molenda et al., 2024; Tang et al., 2024; Wu et al., 2022; Jia et al., 2021). Fingerprinting techniques have recently emerged as another mainstream approach. Instead of modifying the data, they analyze intrinsic properties of the model itself, such as comparing the relative distances between positive and negative samples in the embedding space to determine whether a suspicious model resembles the owner's model closely enough to claim ownership. However, fingerprinting faces a major limitation in that it requires training a large number of positive and negative samples in order to establish the relative relationships necessary for ownership verification (You et al., 2024; Waheed et al., 2024; Cao et al., 2021). In addition, both of the aforementioned approaches encounter a significant challenge: the lack of rigorous theoretical guarantees, which makes them difficult to adopt in real-world production settings. As a result, the development of a certified ownership verification against MEAs remains nascent.

Despite its significance, achieving a certified ownership verification against MEAs remains highly challenging. (1) *Quantifying the similarity of DNNs.* Measuring the similarity or functional equivalence between two models is inherently difficult, as suspicious models may differ in architecture, parameters, or training data, but still exhibit highly similar functionality to the target model. (2) *Defining an effective criterion.* Determining whether a suspicious model has been extracted from a target model via an MEA is inherently difficult, as one typically needs to compare the relative relationships between DNNs to assess whether the suspicious model is closer to the target. While having a threshold that could distinguish such models would be highly desirable, identifying a threshold that reliably separates surrogate models from independent ones remains nontrivial, and alternative strategies for capturing these relationships are often even more complex. (3) *Certification for verification.* Even if we are able to quantify relationships between DNNs and define a corresponding ownership verification scheme, this is because achieving rigorous accuracy guarantees throughout the ownership verification process is highly challenging, and without such guarantees, practical applicability is severely limited. For these reasons, developing a certified ownership verification against MEAs is a non-trivial task. In this paper, we propose CeRtifiED Ownership Verification of Deep Neural Networks against Model ExtractIon ATtacks (**CREDIT**), a certified ownership verification framework designed to protect against MEAs. Specifically, we employ mutual information on the embeddings of DNNs to quantify the relationship between models. We further construct a CREDIT threshold that provides an effective decision criterion for determining whether a model has been obtained through an MEA. Finally, we present rigorous mathematical proofs to certify our defense mechanism. Moreover, we extensively evaluate our method on a wide range of datasets, and the results demonstrate that our approach not only provides a theoretically certified ownership verification but also achieves state-of-the-art performance across all experiments. Our contributions can be summarized in threefolds:

- **Certified Ownership Verification against MEAs.** To the best of our knowledge, this is the first work to propose a certified ownership verification against MEAs. We formally formulate the problem of certified ownership verification for MEAs and introduce the CREDIT method, which enables practical ownership verification while providing rigorous theoretical guarantees.

- **CREDIT Threshold for Ownership Verification.** We propose the novel use of mutual information as a rigorous metric to quantify the dependency between two models, and further introduce the CREDIT threshold as a theoretically principled and practically effective criterion for ownership verification. Moreover, we establish strict probabilistic bounds on the errors induced by this threshold, thereby providing strong theoretical guarantees.

- **Extensive Empirical Evaluation.** We conduct a comprehensive evaluation of the proposed CREDIT method across multiple data modalities and with diverse backbone models to demonstrate its generalization ability. In thorough comparisons with a wide range of baselines, CREDIT consistently achieves state-of-the-art performance.

## 2 PRELIMINARIES

**Notations.** We use lowercase bold letters (e.g., $x$) to denote vectors, uppercase bold letters (e.g., $X$) to denote matrices or collections of samples, and calligraphic letters (e.g., $\mathcal{X}, \mathcal{V}$) to represent sets such as datasets or verification subsets. Scalar values are denoted by lowercase italic letters (e.g., $\tau, \sigma$), and functions are expressed as lowercase italic letters such as $f$ or $g$. We consider a general input domain $\mathcal{X}$, where each input $x \in \mathcal{X}$ may correspond to an image, a text sequence, or a time series instance. A model $f : \mathcal{X} \rightarrow \mathcal{Y}$ maps an input to a target output space $\mathcal{Y}$, and is often equipped with an embedding extractor $e_f : \mathcal{X} \rightarrow \mathbb{R}^d$, which maps inputs to latent representations in $\mathbb{R}^d$. These embeddings are typically obtained from an intermediate or final hidden layer of the model and are used for downstream tasks such as classification.

**Model Extraction Attacks.** We denote the *target model* by $f$, which is trained and owned by the legitimate model owner. The *defense model* is denoted by $g$, which corresponds to the target model $f$ equipped with our designed defense mechanism. We use $h$ to denote a *suspicious model*, which may take one of two forms: $h_{\text{ind}}$ refers to an *independent model* trained by another model owner entirely from scratch using unrelated datasets and potentially different architectures, whereas $h_{\text{sur}}$ denotes a *surrogate model* obtained by an adversary through MEAs on $g$. We will subsequently use these notations to formally define our problem.

**Mutual Information.** The mutual information (MI) (Kraskov et al., 2004; Belghazi et al., 2018) of two random variables is a measure of the mutual dependence between the two variables. Let $X \sim P_X$ be a random input sampled from the data distribution. The mutual information between two random variables $U$ and $V$ is defined as

$$I(U; V) = \mathbb{E}_{(U,V)} \left[ \log \frac{p(U, V)}{p(U)p(V)} \right], \tag{1}$$

which quantifies the amount of information shared between $U$ and $V$. In practice, since the inputs and outputs of deep neural networks generally lack a well-defined distribution, mutual information cannot be computed directly. Instead, we employ an appropriate estimator to approximate it. Specifically, we adopt the KSG estimator (Kraskov et al., 2004), which leverages nearest-neighbor statistics to approximate the underlying probability densities. The detailed computation procedure is provided in Appendix A.1.

**Gaussian Mechanism.** The Gaussian mechanism is a fundamental approach for achieving differential privacy (Dwork, 2006; Abadi et al., 2016). It operates by perturbing the true output of a computation with random noise sampled from a Gaussian distribution, where the magnitude of the noise is carefully determined by the desired privacy parameters. By doing so, it ensures that the contribution of any single data point remains indistinguishable, thereby safeguarding individual privacy while preserving the utility of the released results. Building upon this key theoretical foundation, our defense mechanism is constructed, and we provide its rigorous definition in Appendix B.1.

### 2.1 PROBLEM FORMULATION

In this subsection, we formally present the problem formulation of *certified ownership verification Against Model Extraction Attacks*. Our primary focus is on characterizing the entire defense workflow under the ownership verification setting and specifying the aspects that require certification.

**Definition 1** (Certified Ownership Verification Against MEA)**.** *Let $f$ be the target model and $g$ its defended version. Consider any model $h$, which may be independently trained or obtained through a bounded number of queries to $g$. Given a verification set $\mathcal{V} \subset \mathcal{X}$, let $M\big(e_h(X), e_g(X)\big)$ denote a similarity measure quantifying the relationship between the embeddings of $h$ and $g$ on $\mathcal{V}$. We say that $g$ achieves certified ownership verification against model extraction if there exists a threshold $\tau > 0$ such that, with error probability at most $\gamma$, the criterion based on $M$ correctly distinguishes surrogate models extracted from $g$ from independently trained models.*

The intuition behind Definition 1 is that an attacker will always attempt to train a surrogate model $h_{\text{sur}}$ to mimic the functionality of the target model $f$. Therefore, if we can employ a metric M to quantify the similarity between DNN models, it becomes possible to identify a threshold $\tau$ that distinguishes whether a suspicious model is a surrogate or an independent model, while also providing rigorous theoretical guarantees on the associated error probabilities.

**Problem 1** (Achieving Certified Ownership Verification Against MEA). *Given a target model $f$, we need to construct a defended model $g$. To determine whether a suspicious model $h$ has been extracted from $g$, we measure the similarity between $h$ and $g$ on a verification set $\mathcal{V}$. Based on this measure, we select a verification threshold $\tau$ and require that the resulting decision achieves an error guarantee $\gamma$, ensuring that both false positives and false negatives are bounded.*

## 3 METHODOLOGY

In this section, we provide a detailed description of our certified ownership verification against MEA. We present the complete workflow of our defense model, beginning with its construction and followed by the design of the CREDIT threshold for ownership verification with rigorous theoretical guarantees. Finally, we discuss how to balance model utility and verification effectiveness in practical applications, offering key insights into this trade-off.

### 3.1 BUILDING DEFENSE MODEL VIA A GAUSSIAN MECHANISM

To effectively defend against MEA, it is crucial to first analyze the nature of the attack. Given its similarity to knowledge distillation, the surrogate model $h_{\text{sur}}$ is designed to approximate the output distribution of the target model $f$ with high fidelity. Prior studies (Waheed et al., 2024) have highlighted this phenomenon, noting that the embedding distribution of $h_{\text{sur}}$ often exhibits a high degree of similarity to that of $f$. However, the question remains: *how similar are they, and how can such similarity be quantified?* Naturally, we turn to mutual information as a principled measure of this relationship, as it is capable of capturing both linear and nonlinear dependencies while offering a rigorous information-theoretic interpretation of similarity between random variables (Kraskov et al., 2004; Belghazi et al., 2018). Specifically, we adopt the KSG estimator to measure the similarity between the embeddings of two models. Subsequently, we employ the Gaussian mechanism to our defense model and, building upon it, derive a theoretically guaranteed upper bound on the mutual information. Concretely, we inject independent Gaussian noise $Z \sim \mathcal{N}(0, \sigma^2 I_d)$ into the embeddings of the target model, yielding a defended model $g$. With this mechanism in place, the defended embeddings admit a theoretical upper bound on the mutual information with respect to any other model (Bun & Steinke, 2016).

**Theorem 1** (Mutual Information Bound). *Let $X = (X_1, \ldots, X_n) \sim P_X^n$ be a collection of $n$ independent entries, and let $f : \mathcal{X} \to \mathbb{R}^d$ be a function with global $\ell_2$ sensitivity $\Delta = \sup_{x,x'} \|e_f(x) - e_f(x')\|_2$, where $x$ and $x'$ denote neighboring datapoints. Consider the Gaussian mechanism defined by $e_g(x) = e_f(x) + Z$, where the noise $Z \sim \mathcal{N}(0, \sigma^2 I_d)$ is independent of $X$. Define $\beta = \frac{\Delta^2}{2\sigma^2} n$. For any possibly randomized model with embedding function $e : \mathcal{X} \to \mathbb{R}^d$ such that $e(X) \perp\!\!\!\perp Z \mid X$, we have*

$$I\big(e(X); e_g(X)\big) \leq \beta.$$

The existence of this upper bound is crucial for distinguishing whether a suspicious model originates from a model extraction attack on the protected model. Independent models, trained without knowledge of the target, typically exhibit distributions that diverge significantly from the defended model, resulting in low mutual information. In contrast, the surrogate model $h_{\text{sur}}$, which directly fits the defended model $g$, inherits an embedding distribution highly similar to that of $g$, leading to a high mutual information value. The proof is provided in the Appendix B.1. To further strengthen this result, we establish the tightness of the bound.

**Theorem 2** (Tightness). *Fix $\Delta > 0$ and $\sigma > 0$, and set $\beta = \frac{\Delta^2}{2\sigma^2} n$. There exist a distribution $P_X$ supported on two neighboring inputs, a function $f$ with global $\ell_2$ sensitivity $\Delta$, a Gaussian mechanism $e_g(x) = e_f(x) + Z$ with $Z \sim \mathcal{N}(0, \sigma^2 I_d)$, and a randomized model with embedding function $e$ such that*

$$I\big(e(X)\,;\,e_g(X)\big) \;\geq\; \beta - o(1) \qquad \text{as } \beta \to 0.$$

*Consequently, the upper bound $I(e(X); e_g(X)) \leq \beta$ in Theorem 1 is information theoretically tight.*

Combining these properties, we can theoretically determine a threshold within the range $[0, \beta]$ that effectively separates the surrogate model $h_{\text{sur}}$ from independently trained models $h_{\text{ind}}$. The full proof is provided in the Appendix B.2. In the following subsection, we present the design of our CREDIT threshold and show how it is equipped with theoretical guarantees on error probabilities.

## 3.2 CERTIFIED VERIFICATION THRESHOLD

Prior studies (Waheed et al., 2024) have observed that the embedding distribution of the surrogate model $h_{\mathrm{sur}}$ tends to resemble that of the target model $f$. However, the key question is: *to what extent does this similarity certify that $h_{\mathrm{sur}}$ has been extracted from the protected model?* To address this, we introduce CREDIT threshold $\tau$ for ownership verification of suspicious models.

**Definition 2** (CREDIT Threshold for Ownership Verification). *Let $e_g$ be a defended model whose outputs are protected by Gaussian mechanism, and let $\beta$ denote the mutual information upper bound from Theorem 1. For a verification set $\mathcal{V}$ of cardinality $|\mathcal{V}|$, embedding dimension $d$, query budget $Q$, and constants $\rho \in (0, 1)$ and $\eta \in (0, 1)$, the CREDIT threshold is defined as*

$$\tau = \beta\Big[1 - \rho \exp\big(-Q\,\beta/(\eta\,d\,|\mathcal{V}|)\big)\Big].$$

As mentioned earlier, any suspicious model $h$ has an upper bound $\beta$ on its mutual information with our defense model $g$. Our task is to determine the CREDIT threshold within the interval $[0, \beta]$. This threshold is influenced by multiple factors. For surrogate models, the query budget $Q$ largely determines the degree to which the functionality of the target model can be duplicated. For the verification process, the size of the verification set $\mathcal{V}$ affects the accuracy of the mutual information estimation. In addition, the embedding dimension d used for MI estimation also improves the precision of the estimate. The intuition behind Definition 2 is that when $Q$ is larger, the surrogate model more closely replicates the functionality of the target, and thus the threshold should be higher. Conversely, as the verification set $\mathcal{V}$ grows and the embedding dimension d increases, the MI estimation becomes more accurate, and therefore a lower threshold is sufficient. We then establish error bounds for the CREDIT threshold when it is applied to ownership verification.

**Theorem 3** (Certified Ownership Verification Guarantee). *Let $\widehat{I}$ denote the empirical mutual information on a verification set $\mathcal{V}$, and let $\tau$ be the threshold of Definition 2. Then the following hold: (1) Independent False Alarm (Type I error): For any independently trained model $h_{\mathrm{ind}}$,*

$$\Pr\Big[\widehat{I}\big(e_{h_{\mathrm{ind}}}(X), e_g(X)\big) > \tau\Big] \;\leq\; \exp\Big(-\frac{2|\mathcal{V}|\,\tau^2}{C^2}\Big) \;\triangleq\; \gamma_1.$$

*(2) Surrogate Missed Detection (Type II error): For any surrogate model $h_{\mathrm{sur}}$,*

$$\Pr\Big[\widehat{I}\big(e_{h_{\mathrm{sur}}}(X), e_g(X)\big) \leq \tau\Big] \;\leq\; \exp\Big(-\frac{2|\mathcal{V}|\,\big(I(e_{h_{\mathrm{sur}}}(X), e_g(X))-\tau\big)^2}{C^2}\Big) \;\triangleq\; \gamma_2.$$

*Both $\gamma_1$ and $\gamma_2$ decay exponentially in the verification set size $|\mathcal{V}|$, so enlarging $\mathcal{V}$ drives the error probabilities below any desired tolerance. Here $C$ is the bounded difference constant of $\widehat{I}$.*

In summary, our CREDIT threshold $\tau$ provides rigorous guarantees against both Independent False Alarm (Type I error) and Surrogate Missed Detection (Type II error) probabilities. Full proofs are provided in the Appendix B.3.

## 3.3 PRACTICAL TRADE-OFF: UTILITY VS. VERIFICATION EFFECTIVENESS

When applying our proposed CREDIT framework to defend the target model, there exists an inherent trade-off between preserving the utility of the defended model on downstream tasks and enhancing the effectiveness of ownership verification. In this section, we investigate how this trade-off relates to the Gaussian perturbation introduced in our defense model, and we ultimately show how to optimize the perturbation to achieve the best balance between the two objectives.

**Utility Performance.** We first aim to establish the connection between Gaussian perturbation and model utility. Since the exact distribution of the clean embeddings is difficult to characterize, we use a reasonable approximation: a zero-mean Gaussian with covariance $\Sigma$. Then, based on the maximum entropy property of Gaussian distribution, we define the *utility entropy gain* as $\Delta H_{\mathrm{util}}(\sigma) := H_{\mathrm{G}}(X + Z) - H_{\mathrm{G}}(X)$, where $Z \sim \mathcal{N}(0, \sigma^2 I)$. A larger $\Delta H_{\mathrm{util}}(\sigma)$ indicates that the perturbed embeddings deviate more strongly from the information content of the clean distribution, approaching the behavior of pure Gaussian noise. Consequently, the preserved utility decreases as $\Delta H_{\mathrm{util}}(\sigma)$ grows, whereas smaller values of $\Delta H_{\mathrm{util}}(\sigma)$ correspond to higher utility preservation. We provide detailed demonstration in Appendix A.2

**Verification Effectiveness.** We further assess verification effectiveness, formulated as a binary hypothesis test that distinguishes an independent model $h_{\text{ind}}$ from a surrogate model $h_{\text{sur}}$ trained with at most $Q$ queries. The verifier computes an empirical mutual information statistic $\widehat{I}$ and compares it against a threshold $\tau$. To capture the residual uncertainty of this decision, we introduce the *verification entropy*: $\mathcal{H}_{\text{ver}}(\sigma) := H(T_\sigma \mid H)$, where $T_\sigma$ is the verification indicator and $H$ is the true hypothesis. Intuitively, $\mathcal{H}_{\text{ver}}$ measures the verification ambiguity: it is zero when the test is always correct, and increases as the probabilities of false positives or false negatives grow. Consequently, smaller values of the *verification entropy gain* indicate stronger robustness. We provide a detailed demonstration in Appendix A.3.

**Sigma Selection.** The noise parameter $\sigma$ simultaneously governs the trade-off between utility and verification robustness. The utility entropy cost $\Delta H_{\text{util}}(\sigma)$ grows with $\sigma$, while the verification entropy $\Delta H_{\text{ver}}(\sigma)$ decreases. Thus, choosing $\sigma$ reduces to the following minimization problem:

$$\min_\sigma \quad \lambda_{\text{util}} \, \Delta H_{\text{util}}(\sigma) + \lambda_{\text{ver}} \, \Delta H_{\text{ver}}(\sigma),$$

where $\lambda_{\text{util}}$ and $\lambda_{\text{ver}}$ are tunable weights controlling the relative emphasis on utility preservation versus verification robustness. In practice, we first select a candidate range of $\sigma$ values and perform a grid search to identify the optimal choice. For the utility term $\Delta H_{\text{util}}(\sigma)$, we empirically sample embeddings from the target model, estimate their covariance spectrum, and then evaluate the entropy increase. For the verification term $\Delta H_{\text{ver}}(\sigma)$, we directly compute the corresponding error rates $\gamma_1$, $\gamma_2$ induced by each $\sigma$, and substitute them into the binary entropy expression. The overall objective function then yields the optimal $\sigma$. We provide additional empirical results in the Appendix C.5, where we show how the $\sigma$ obtained from this procedure improves the trade-off between utility preservation and verification robustness.

## 4 EXPERIMENT

In this section, we present a comprehensive evaluation of CREDIT. Specifically, we aim to address three research questions: **RQ1:** How effective is our proposed CREDIT in simultaneously preserving model utility and ensuring verification effectiveness against surrogate models? **RQ2:** How does our proposed method improve efficiency compared to baseline methods? **RQ3:** How do different parameter choices affect the reliability of ownership verification certification?

### 4.1 EXPERIMENTAL SETUP

**Datasets.** Our experiments primarily focus on the image classification task. We adopt widely used datasets across different data modalities, including CIFAR-10 (Krizhevsky et al., 2009) and CIFAR-100 (Krizhevsky et al., 2009) in Computer Vision domain, as well as ENZYMES (Morris et al., 2020) and PROTEINS (Morris et al., 2020) in Graph Learning domain. For each dataset, we split the training and test sets accordingly. In particular, we enforce a strict separation between the query set and the verification set to ensure no overlap. The detailed dataset statistics and the exact splitting strategy are provided in the Appendix C.1.

**Backbone Models.** We mainly use ResNet-50 (He et al., 2016), VGG-16 (Simonyan & Zisserman, 2014), DenseNet (Huang et al., 2017), and GoogLeNet (Szegedy et al., 2015) as backbone models in image classification task, and GCN (Kipf & Welling, 2016), GAT (Veličković et al., 2017), GraphSAGE (Hamilton et al., 2017) and SSGC (Zhu & Koniusz, 2021) as backbone models in graph classification task. By training different backbones with various optimization strategies, we obtain a large number of surrogate models and independent models to support verification evaluation. During the attack phase, we adopt the widely accepted Knowledge Distillation (Romero et al., 2014) method as the attack strategy. To ensure consistency with prior work, we strictly follow the query strategy introduced in KnockOff (Orekondy et al., 2019a).

**Baselines.** We propose CREDIT as a defense method against Model Extraction Attacks (MEAs) under the ownership verification setting. Defense strategies in this setting can be broadly categorized into watermarking and fingerprinting. Specifically, in the computer vision domain we adopt EWE (Jia et al., 2021), Backdoor (Adi et al., 2018), IPGuard (Cao et al., 2021), and UAP (Peng et al., 2022) as baselines, while in the graph domain we consider RandomWM (Zhao et al., 2021),

BackdoorWM (Xu et al., 2023), SurviveWM (Wang et al., 2023), and ImperceptibleWM (Zhang et al., 2024b). The implementation details of all baselines are provided in the Appendix C.3.

**Metrics.** To comprehensively evaluate CREDIT, we consider multiple aspects of performance. For model utility and verification robustness, we primarily measure Accuracy and AUROC, respectively. For model efficiency, we evaluate the preparation and verification time introduced by the defense mechanism. For certification reliability, we analyze the interplay of different parameter settings and their impact on model utility. We will provide a detailed analysis in the following subsections.

Table 1: Evaluation of all defense methods on downstream tasks in terms of utility performance. All results are reported as accuracy, with the best performance highlighted in **bold**.

| Dataset | Backbone | Vanilla | Backdooring | EWE | IPGuard | UAP | **CREDIT** |
|---------|----------|---------|-------------|-----|---------|-----|------------|
| CIFAR-10 | ResNet | $95.70 \pm 0.03$ | $78.10 \pm 2.18$ | $90.25 \pm 0.02$ | $91.08 \pm 0.00$ | $84.73 \pm 2.26$ | $\mathbf{94.67 \pm 0.22}$ |
| | VGG | $92.18 \pm 0.01$ | $77.60 \pm 2.76$ | $88.82 \pm 0.03$ | $89.80 \pm 0.02$ | $82.91 \pm 2.63$ | $\mathbf{91.68 \pm 0.17}$ |
| | DenseNet | $93.33 \pm 0.07$ | $60.45 \pm 3.67$ | $86.22 \pm 0.10$ | $89.41 \pm 0.00$ | $74.82 \pm 4.50$ | $\mathbf{92.58 \pm 0.15}$ |
| | GoogLeNet | $94.90 \pm 0.04$ | $89.31 \pm 1.38$ | $93.58 \pm 0.03$ | $92.98 \pm 0.19$ | $91.84 \pm 0.92$ | $\mathbf{94.67 \pm 0.11}$ |
| CIFAR-100 | ResNet | $80.48 \pm 0.11$ | $74.43 \pm 0.17$ | $75.85 \pm 0.07$ | $77.54 \pm 0.05$ | $64.30 \pm 4.14$ | $\mathbf{79.90 \pm 0.21}$ |
| | VGG | $77.41 \pm 0.09$ | $73.76 \pm 0.29$ | $74.48 \pm 0.03$ | $76.22 \pm 0.01$ | $71.10 \pm 1.58$ | $\mathbf{76.71 \pm 0.20}$ |
| | DenseNet | $80.34 \pm 0.14$ | $72.91 \pm 0.46$ | $76.72 \pm 0.08$ | $77.95 \pm 0.03$ | $57.25 \pm 5.02$ | $\mathbf{79.29 \pm 0.27}$ |
| | GoogLeNet | $78.72 \pm 0.10$ | $71.49 \pm 0.33$ | $74.69 \pm 0.04$ | $76.41 \pm 0.01$ | $66.96 \pm 3.12$ | $\mathbf{77.50 \pm 0.19}$ |

| Dataset | Backbone | Vanilla | RandomWM | BackdoorWM | SurviveWM | ImperceptibleWM | **CREDIT** |
|---------|----------|---------|-----------|------------|-----------|-----------------|------------|
| ENZYMES | GCN | $42.78 \pm 2.58$ | $39.11 \pm 2.58$ | $38.16 \pm 3.79$ | $15.27 \pm 3.36$ | $26.94 \pm 7.83$ | $\mathbf{40.61 \pm 1.46}$ |
| | GAT | $41.94 \pm 1.42$ | $36.94 \pm 2.19$ | $35.83 \pm 1.80$ | $18.33 \pm 1.18$ | $22.78 \pm 4.16$ | $\mathbf{38.56 \pm 1.42}$ |
| | GraphSAGE | $50.83 \pm 5.57$ | $42.22 \pm 5.50$ | $43.61 \pm 4.78$ | $10.27 \pm 1.04$ | $27.78 \pm 5.67$ | $\mathbf{46.94 \pm 5.54}$ |
| | SSGC | $33.05 \pm 3.07$ | $28.61 \pm 2.71$ | $28.27 \pm 3.14$ | $16.39 \pm 1.04$ | $26.17 \pm 3.79$ | $\mathbf{28.94 \pm 2.19}$ |
| PROTEINS | GCN | $71.00 \pm 2.60$ | $70.40 \pm 1.84$ | $70.24 \pm 0.92$ | $59.49 \pm 3.07$ | $66.37 \pm 2.64$ | $\mathbf{72.50 \pm 1.12}$ |
| | GAT | $73.09 \pm 1.32$ | $72.80 \pm 2.02$ | $72.94 \pm 0.42$ | $40.96 \pm 1.48$ | $53.66 \pm 2.75$ | $\mathbf{74.14 \pm 1.56}$ |
| | GraphSAGE | $74.59 \pm 0.76$ | $70.69 \pm 0.42$ | $70.84 \pm 0.21$ | $38.71 \pm 0.56$ | $52.46 \pm 6.99$ | $\mathbf{73.44 \pm 1.69}$ |
| | SSGC | $73.99 \pm 0.37$ | $71.00 \pm 2.10$ | $73.99 \pm 1.10$ | $47.23 \pm 2.15$ | $71.00 \pm 1.12$ | $\mathbf{74.73 \pm 2.11}$ |

## 4.2 MODEL UTILITY & VERIFICATION EFFECTIVENESS

In this section, we analyze the performance of the proposed CREDIT method compared with existing baselines in terms of both model utility performance and ownership verification effectiveness. Specifically, we evaluate CREDIT across two different data modalities. Leveraging the generalization ability of our approach, we further deploy CREDIT on four distinct backbone models within each modality to implement the ownership verification methods. As shown in Table 1, on

Table 2: Evaluation of defense methods on ownership verification after MEAs. Results are reported as AUC, with the best performance highlighted in **bold**.

| Method | ResNet | VGG | DenseNet | GoogLeNet |
|--------|--------|-----|----------|-----------|
| Backdooring | 58.64 | 51.85 | 74.07 | 80.25 |
| EWE | 51.24 | 62.96 | 48.77 | 62.35 |
| IPGuard | 40.74 | 61.11 | 67.90 | 50.62 |
| UAP | 52.47 | 67.28 | 72.22 | 79.63 |
| **Our** | **100.0** | **100.0** | **100.0** | **100.0** |

image classification tasks, CREDIT achieves the best utility performance on both CIFAR-10 and CIFAR-100, with only negligible degradation. This demonstrates that with reasonable parameter choices, utility can be effectively preserved. On graph classification tasks, CREDIT likewise delivers consistently superior results. Interestingly, on certain datasets, CREDIT even outperforms the vanilla model. This occurs because the representational capacity of some backbone models is limited, and the addition of noise to the embeddings can, in such cases, enhance performance. These results clearly demonstrate that our CREDIT method not only achieves strong utility performance but also exhibits remarkable generalization ability, enabling deployment across different backbone models and even multiple data modalities. We further assess ownership verification effectiveness. Specifically, we train multiple independent models and surrogate models to examine whether ownership can be reliably verified against all suspicious models. Following the strictest model extraction setting C.4, we first deploy the corresponding defense mechanism on the target model, then perform ME attacks on the defense model to obtain surrogate models. In parallel, we construct independent models under entirely different training configurations. As summarized in Table 2, our experiments

on CIFAR-10 show that CREDIT consistently outperforms all baseline methods, clearly demonstrating its superior effectiveness in ownership verification.

## 4.3 EVALUATION OF MODEL EFFICIENCY

Table 3: Efficiency of ownership verification methods in both the preparation and verification stages across backbone models with different parameter sizes. Results are reported in seconds, with the best performance highlighted in **bold**.

| Method | VGG-16 (15M) | | ResNet-50 (25.6M) | |
|---|---|---|---|---|
| | Preparation(s) | Verification(s) | Preparation(s) | Verification(s) |
| Backdooring | $0.594 \pm 0.11$ | $83.10 \pm 0.85$ | $0.399 \pm 0.09$ | $188.8 \pm 0.40$ |
| EWE | $0.279 \pm 0.00$ | $73.40 \pm 0.13$ | $0.199 \pm 0.02$ | $189.0 \pm 0.76$ |
| IPGuard | $1.783 \pm 0.00$ | $73.94 \pm 0.23$ | $4.319 \pm 0.00$ | $189.0 \pm 0.41$ |
| UAP | $5.587 \pm 0.13$ | $72.74 \pm 0.08$ | $5.722 \pm 0.14$ | $180.3 \pm 0.32$ |
| **Our** | $\mathbf{0.001 \pm 0.00}$ | $\mathbf{22.23 \pm 0.07}$ | $\mathbf{0.001 \pm 0.00}$ | $\mathbf{22.62 \pm 0.36}$ |

We next evaluate the efficiency of CREDIT. We divide the entire ownership verification process of each defense model into two stages. The first stage is the preparation stage, which measures the time required for each model to execute its corresponding defense mechanism. For baseline methods, this corresponds to constructing watermarks or fingerprints, while for CREDIT, it only involves computing the CREDIT threshold. The second stage is the verification stage, which measures the time required to perform a complete ownership verification. For baselines, regardless of whether they rely on watermarks or fingerprints, this requires training auxiliary models to support the verification process, then querying the suspicious model, comparing the responses with the watermark or fingerprint, and finally combining the auxiliary model's results to reach a decision. In contrast, CREDIT only requires estimating the mutual information between the embeddings of the suspicious model and the defense model during inference, followed by a direct comparison with the CREDIT threshold to obtain the final decision. As shown in Table 3, CREDIT demonstrates outstanding time efficiency. Since the preparation stage only involves computing $\tau$, it incurs minimal overhead. Moreover, during verification, CREDIT requires no auxiliary model training, giving it a significant advantage over all existing baselines. Additionally, as the backbone model size increases, baseline methods suffer from additional computational costs, whereas CREDIT remains unaffected, as it only relies on estimating mutual information between DNNs. These results highlight the superior efficiency of CREDIT.

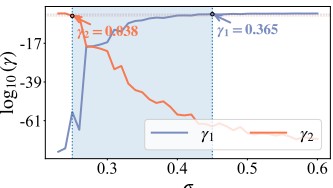 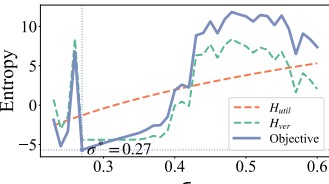 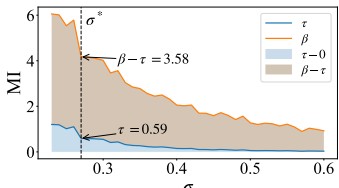

(a) The impact of $\sigma$ on Independent False Alarm and Surrogate Missed Detection probabilities.

(b) Optimizing $\sigma$ through the trade-off between utility and verification effectiveness.

(c) The effect of varying $\sigma$ on threshold $\tau$ and mutual information upper-bound $\beta$.

Figure 1: Analysis of the reliability of certification under different choices of $\sigma$.

## 4.4 CERTIFICATION RELIABILITY: IMPACT OF PARAMETER CHOICES

Finally, we analyze the reliability of certification under different parameter settings. The parameter $\sigma$ plays a central role in our defense model, as it directly determines the upper bound of mutual information between DNNs. At the same time, it fixes the CREDIT threshold $\tau$ for ownership verification, from which we can also derive rigorous guarantees on the two types of error probabilities. In particular, by varying $\sigma$ and computing the two error probabilities, we observe in Figure 1a that as $\sigma$ increases, $\gamma_1$ rises, indicating a higher probability of Independent False Alarm, while $\gamma_2$ decreases, corresponding to a lower probability of Surrogate Missed Detection. We further identify a reasonable tolerance range, where $\sigma$ between 0.25 and 0.45 yields acceptable performance. Within this range, the probability of Surrogate Missed Detection persists at a low rate, ensuring stricter and more reliable detection of surrogate models. As discussed in Section 3.3, we formulate utility and verification effectiveness as a joint optimization problem. By optimizing this objective, we obtain the theoretically optimal value of $\sigma$. As illustrated in Figure 1b, this optimum occurs at $\sigma = 0.27$.

Once $\sigma$ is fixed, both the upper bound $\beta$ and the CREDIT threshold $\tau$ are uniquely determined, as shown in Figure 1c. In this case, the addition of noise ensures that the mutual information with an independent model remains negligible, while any suspicious model whose mutual information falls within the broad interval between $\tau$ and $\beta$ can be confidently identified as a surrogate.

## 5 RELATED WORK

**Model Extraction Attacks & Defenses.** Model Extraction Attacks (MEAs) and Model Extraction Defenses (MEDs) have recently attracted significant attention, driven by the increasing popularity of the MLaaS deployment paradigm. MEAs demonstrate the inherent vulnerability of MLaaS (Li et al., 2024), as adversaries can often obtain functionality-equivalent models at very low cost. For example, Knockoff (Orekondy et al., 2019a) proposed a two-step procedure in which the adversary first queries the target model and then trains a knockoff model on the collected input–output pairs, achieving reasonable performance with minimal expense. ADBA (Wang et al., 2025) introduced Approximation Decision Boundary Analysis to effectively attack target models by exploiting decision boundary information. In the graph domain, CEGA (Wang et al.) showed that selecting informative nodes can drastically reduce the query budget while enabling cost-effective model extraction with strong performance. On the defense side, MEDs have also been extensively studied. Fingerprinting methods (Cao et al., 2021; Peng et al., 2022; You et al., 2024; Waheed et al., 2024) aim to identify intrinsic model characteristics and verify ownership by checking whether a suspicious model exhibits these characteristics. Watermarking methods (Jia et al., 2021; Adi et al., 2018; Zhao et al., 2021; Xu et al., 2023) instead embed specially designed watermark samples into the target model, such that ownership can be claimed if these samples can be reliably verified on a suspicious model. Perturbation-based defenses (Lee et al., 2018; Kariyappa & Qureshi, 2020; Orekondy et al., 2019b) adopt yet another strategy, injecting noise into outputs to prevent adversaries from extracting usable surrogate models. Different from all the above approaches, our work is the first to introduce a certified ownership verification against MEAs, providing theoretical guarantees for practical ownership verification.

**Certified Defense for Deep Learning Models.** Certified defenses for deep neural networks (DNNs) have been extensively studied, owing to their theoretical guarantees and practical significance. In robustness (Zheng et al., 2016; Katz et al., 2017), the goal is to analyze how well DNNs retain effectiveness when subjected to perturbations. Several works have addressed this direction: one line of research (Cohen et al., 2019) leverages randomized smoothing to obtain certified accuracies, while another (Lecuyer et al., 2019) achieves certified robustness against adversarial examples through differential privacy. Beyond robustness, certified unlearning (Bourtoule et al., 2021; Nguyen et al., 2025) has emerged as an important topic as privacy concerns continue to grow and users increasingly demand the removal of specific data from models. For instance, one work (Dong et al., 2024) proposed flexible and certified unlearning for graph neural networks (GNNs) (Kipf & Welling, 2016; Veličković et al., 2017; Hamilton et al., 2017), addressing four different types of unlearning requests with rigorous theoretical guarantees. Another (Zhang et al., 2024a) introduced certified unlearning methods for DNNs, which are applicable to nonconvex objectives and enable efficient computation through inverse Hessian approximation. Certification has also been extended to intellectual property protection (Xue et al., 2021; Sun et al., 2023; Zhang et al., 2018). IPCert (Jiang et al., 2023) proposed turning existing watermarking and fingerprinting schemes into provably robust methods against model perturbations via randomized smoothing. Distinct from all of the above, our work is the first to propose a certified ownership verification against Model Extraction Attacks (MEAs), thereby addressing a critical gap in ensuring provable protection for DNNs under extraction threats.

## 6 CONCLUSION

In this work, we introduced CREDIT, the first certified defense against Model Extraction Attacks. We formally formulated the problem of certified defense in this setting, emphasizing the necessity of theoretical guarantees for practical deployment. To bridge the gap in defending DNNs against MEAs, CREDIT employs mutual information to systematically quantify the relationship between models and establishes a threshold-based criterion for ownership verification. We further provided rigorous mathematical guarantees on the error bounds associated with this threshold. Extensive

experiments across diverse datasets and modalities demonstrate that CREDIT consistently achieves state-of-the-art performance, highlighting both the effectiveness and generality of our approach.

ETHICS STATEMENT

Our work focuses on developing certified ownership verification against model extraction attacks, which are designed to protect the intellectual property of machine learning models. The datasets used in our experiments are all standard public benchmarks for image and graph classification tasks (e.g., CIFAR-10, CIFAR-100, ENZYMES, PROTEINS), which do not contain sensitive personal information. Therefore, our study does not raise specific ethical concerns regarding privacy or data misuse. We believe our contributions may promote responsible AI development by providing principled tools for safeguarding model ownership.

REPRODUCIBILITY STATEMENT

We have taken concrete steps to ensure the reproducibility of our work. All backbone architectures (for both image and graph classification) are standard implementations directly obtained from `torchvision` and `torch_geometric`. For baseline defenses where official code was not available, we carefully followed the descriptions in the original papers and detailed our re-implementations in Appendix C.3. Hyperparameters, training details, and evaluation protocols are explicitly reported in the main text and appendix. We will release our source code, configuration files, and scripts upon publication to facilitate full reproducibility.

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

# A SUPPLEMENTARY TECHNICAL DETAILS

## A.1 MUTUAL INFORMATION ESTIMATOR

In practice, computing mutual information for high-dimensional embeddings in deep neural networks requires a statistical estimator. We adopt the KSG estimator (Kraskov et al., 2004), which relies on nearest-neighbor statistics to approximate the underlying probability densities:

$$\widehat{I}_{\text{KSG}} = \psi(k) - \frac{1}{n}\sum_{i=1}^{n}\Big[\psi\big(n_x(i)+1\big) + \psi\big(n_y(i)+1\big)\Big] + \psi(n). \tag{2}$$

Here $k$ is the fixed neighborhood size, $\psi$ is the digamma function, and $n_x(i)$ (resp. $n_y(i)$) counts the number of samples, excluding $i$, whose distance in the $Z_1$ (resp. $Z_2$) space is within the joint $k$-NN radius $\varepsilon_i$. Intuitively, the estimator measures how often neighbors in the joint space $(Z_1, Z_2)$ also remain neighbors when projected to the marginal spaces, thereby capturing statistical dependence without explicitly estimating densities. We further assume embeddings are bounded, i.e., $\|Z_{1,i}\|_2 \leq B_1$ and $\|Z_{2,i}\|_2 \leq B_2$, ensuring that distances and neighbor counts are well-defined.

## A.2 DERIVATION OF UTILITY ENTROPY

We provide the derivation of the Gaussian surrogate entropy used in the main content. Consider a random variable $X \in \mathbb{R}^d$ with empirical covariance $\Sigma \in \mathbb{R}^{d \times d}$. Among all continuous distributions with covariance $\Sigma$, the Gaussian distribution maximizes the differential entropy. Therefore, it is natural to approximate the empirical distribution of the embeddings by a zero mean Gaussian $\mathcal{N}(0, \Sigma)$. For a Gaussian random vector $G \sim \mathcal{N}(0, \Sigma)$, the differential entropy is given by

$$H(G) \;=\; \tfrac{1}{2}\log\big((2\pi e)^d \det(\Sigma)\big) = \tfrac{d}{2}\log(2\pi e) + \tfrac{1}{2}\log\det(\Sigma).$$

When isotropic Gaussian noise $Z \sim \mathcal{N}(0, \sigma^2 I)$ is added to $X$, the resulting distribution is $\mathcal{N}(0, \Sigma + \sigma^2 I)$. Its entropy is

$$H_{\text{G}}(X + Z) \;=\; \tfrac{1}{2}\log\big((2\pi e)^d \det(\Sigma + \sigma^2 I)\big).$$

Subtracting the clean entropy $H_{\text{G}}(X)$ yields

$$\Delta H_{\text{util}}(\sigma) = H_{\text{G}}(X + Z) - H_{\text{G}}(X) = \tfrac{1}{2}\log\det\Big(I + \tfrac{\sigma^2 I}{\Sigma}\Big).$$

This expression quantifies the increase in entropy introduced by Gaussian perturbation. Since higher entropy indicates less structure and more randomness, a smaller $\Delta H_{\text{util}}(\sigma)$ corresponds to better preservation of the informative content of the original embeddings.

## A.3 DERIVATION OF VERIFICATION ENTROPY

**Hypothesis testing setup.** The verification problem is cast as a binary hypothesis test:

$$H = \begin{cases} \text{ind}, & \text{if the queried model is independent of the protected model,} \\ \text{sur}, & \text{if the queried model is a surrogate trained with at most } Q \text{ queries.} \end{cases}$$

Given a sample set $\mathcal{S}$ with $|\mathcal{S}|$ queries, the verifier computes the empirical mutual information statistic $\widehat{I}$. The indicator is

$$T_\sigma = \mathbf{1}\{\widehat{I} > \tau(\sigma, Q)\},$$

where $T_\sigma$ is a binary random variable indicating the verifier's choice, and $\tau(\sigma, Q)$ is a threshold that may depend on the noise parameter $\sigma$ and query budget $Q$.

**Error probabilities.** We have two types of error, *Independent False Alarm (Type I error)*: $\Pr[T_\sigma = 1 \mid H = \text{ind}]$, i.e., incorrectly declaring the model a surrogate. *Surrogate Missed Detection (Type II error)*: $\Pr[T_\sigma = 0 \mid H = \text{sur}]$, i.e., failing to detect a surrogate. By concentration bounds for empirical mutual information, these probabilities can be bounded as

$$\Pr[T_\sigma = 1 \mid H = \text{ind}] \leq \gamma_1(\sigma, Q) = \exp\Big(-\tfrac{2|\mathcal{S}|\,\tau(\sigma,Q)^2}{C^2}\Big),$$

$$\Pr[T_\sigma = 0 \mid H = \mathrm{sur}] \leq \gamma_2(\sigma, Q) = \exp\Big(-\frac{2|\mathcal{S}|\left(I_*(\sigma, Q) - \tau(\sigma, Q)\right)^2}{C^2}\Big),$$

where $C$ is a bound on the range of the score function and $I_*(\sigma, Q)$ is the population mutual information under the surrogate model.

**Verification entropy.** Let $\pi_0 = \Pr[H = \mathrm{ind}]$ and $\pi_1 = \Pr[H = \mathrm{sur}]$ denote the prior probabilities. The conditional entropy of the decision $T_\sigma$ given $H$ is

$$\mathcal{H}_{\mathrm{ver}}(\sigma) = \pi_0\, h_b\big(\Pr[T_\sigma = 1 \mid H = \mathrm{ind}]\big) + \pi_1\, h_b\big(\Pr[T_\sigma = 0 \mid H = \mathrm{sur}]\big),$$

$$\mathcal{H}_{\mathrm{ver}}(\sigma) = \pi_0\, h_b\big(\gamma_1(\sigma, Q)\big) + \pi_1\, h_b\big(\gamma_2(\sigma, Q)\big),$$

where $h_b(p) = -p \log p - (1-p) \log(1-p)$ is the binary entropy function. In practice, if no prior knowledge about $H$ is available, it is common to assume a uniform prior $\pi_0 = \pi_1 = 0.5$. Under the noiseless setting ($\sigma = 0$), the test achieves perfect accuracy, so $\mathcal{H}_{\mathrm{ver}}(0) = 0$. The *verification entropy gain* is then

$$\Delta H_{\mathrm{ver}}(\sigma) = \mathcal{H}_{\mathrm{ver}}(\sigma).$$

This formulation makes clear that the entropy grows as error probabilities increase, quantifying the loss of verification robustness introduced by Gaussian perturbation.

# B  PROOFS

## B.1  PROOF OF THEOREM 1

**Definition 3** (Gaussian Mechanism). *Let $f : \mathcal{D} \to \mathbb{R}^d$ be a query and let $\sigma > 0$ be a noise scale parameter. The* Gaussian mechanism $e_g$ *is defined as*

$$e_g(X) = f(X) + Z, \qquad \text{where } Z \sim \mathcal{N}\big(0, \sigma^2 I_d\big).$$

*This mechanism provides differential privacy based on the global $\ell_2$ sensitivity of the query, defined as $\Delta_2 = \sup_{x,x'} \|f(x) - f(x')\|_2$ To ensure that $e_g$ satisfies $(\epsilon, \delta)$-differential privacy for given $\epsilon \in (0, 1)$ and $\delta \in (0, 1)$, the noise scale $\sigma$ must be chosen to meet the following condition:*

$$\sigma \geq \frac{\sqrt{2\ln\big(1.25/\delta\big)}\,\Delta_2}{\epsilon}.$$

**Theorem 4** (Mutual Information Bound for the Gaussian Mechanism). *Let $f : \mathcal{X} \to \mathbb{R}^d$ be a function with global $\ell_2$ sensitivity $\Delta_2$. Consider the Gaussian mechanism $e_g(X) = f(X) + Z$, where the noise $Z \sim \mathcal{N}(0, \sigma^2 I_d)$ is independent of the input $X$. The mutual information between the input $X$ and the output $e_g(X)$ is bounded as follows:*

$$I(X; e_g(X)) \leq \frac{\Delta^2}{2\sigma^2} n.$$

**Definition 4** (Data Processing Inequality, DPI). *For any random variables forming a Markov chain $X \to Y \to Z$, which implies that $X$ and $Z$ are conditionally independent given $Y$, the mutual information satisfies:*

$$I(X; Z) \leq I(X; Y).$$

**Lemma 5** (Markov Chain from a Common Ancestor). *Let $X$ be a random variable. Let $Y = e_h(X)$ and $Z = e_g(X)$ be two random variables generated from $X$ via two (possibly randomized) functions, $e_h$ and $e_g$. Then, the random variables $Y, X, Z$ form a Markov chain $Y \to X \to Z$.*

*Proof of Lemma 5.* To prove that $Y \to X \to Z$ is a Markov chain, we must show that $Y$ and $Z$ are conditionally independent given $X$. That is, $P(Y, Z|X = x) = P(Y|X = x)P(Z|X = x)$ for all $x$. By definition, the generation of $Y$ depends only on $X$ (via mechanism $e_h$), and the generation of $Z$ depends only on $X$ (via mechanism $e_g$). Once the value of $X$ is fixed, the two generation processes are independent of each other. Therefore, their conditional joint probability factors into the product of their conditional marginal probabilities, establishing the Markov chain. $\square$

**Corollary 6** (Mutual Information Bound Between Two Privatized Outputs). *Let $X$ be a random variable, and let $Y = e_h(X)$ and $Z = e_g(X)$. If the $e_g$ is the Gaussian mechanism, then the mutual information between $Y$ and $Z$ is bounded by:*

$$I(Y; Z) \leq \beta.$$

*Proof of Theorem 1.* From Lemma 5, we have established that the random variables form the Markov chain $Y \to X \to Z$.

By the Data Processing Inequality (Definition 4), this Markov chain implies:

$$I(e_h(X); e_g(X)) \leq I(X; e_g(X)).$$

From Theorem 4, since $e_g$ is gaussian mechanism, we have the bound:

$$I(X; e_g(X)) \leq \beta.$$

Combining these two inequalities yields the desired result in Theorem 1:

$$I(e_h(X); e_g(X)) \leq \beta.$$

$\square$

### B.2 PROOF OF THEOREM 2

*Proof of Theorem 2.* **Construction.** Let $\mathcal{X} = \{x_0, x_1\}$ with $P_X(x_0) = P_X(x_1) = \frac{1}{2}$. Define $f(x_0) = -\Delta_2/2$ and $f(x_1) = +\Delta_2/2$, so that $\|f(x_0) - f(x_1)\|_2 = \Delta_2$. Let $e_g(x) = f(x) + Z$ with $Z \sim \mathcal{N}(0, \sigma^2)$ independent of $X$. Choose $e_h(X) = X$, hence $I(e_h(X); e_g(X)) = I(X; e_g(X))$.

**Binary Gaussian channel form.** Then

$$e_g(X) \sim \begin{cases} \mathcal{N}(-\Delta_2/2, \ \sigma^2) & \text{w.p. } \frac{1}{2}, \\ \mathcal{N}(+\Delta_2/2, \ \sigma^2) & \text{w.p. } \frac{1}{2}. \end{cases}$$

Define the parameter $s \triangleq 1/\sigma^2$. Using the standard normalization, write $Y = f(X) + Z$ with $Z \sim \mathcal{N}(0, \sigma^2)$, so by scaling, $\tilde{Y} = \sqrt{s}\, f(X) + N$ with $N \sim \mathcal{N}(0, 1)$.

**Lower bound via the $I$–MMSE relation.** For any real $X$, $I(X; \sqrt{t}\, X + N) = \frac{1}{2} \int_0^t \text{mmse}(u)\, du$ Guo et al. (2005). Take $t = s$ and note $\text{Var}(f(X)) = \Delta_2^2/4$ for our symmetric binary $X$. It is known that $\text{mmse}(u) = \text{Var}(f(X)) - O(u)$ as $u \downarrow 0$; hence there exists $C_0 > 0$ with $\text{mmse}(u) \geq \text{Var}(f(X)) - C_0 u$ for all small $u$. Thus

$$I(X; e_g(X)) \ = \ \frac{1}{2} \int_0^s \text{mmse}(u)\, du \ \geq \ \frac{1}{2} \int_0^s \left( \frac{\Delta_2^2}{4} - C_0 u \right) du \ = \ \frac{\Delta_2^2}{8}\, s \ - \ \frac{C_0}{4}\, s^2.$$

Substitute $s = 1/\sigma^2$ and $\beta = \Delta_2^2/(2\sigma^2)$ to obtain

$$\frac{\Delta_2^2}{8}\, s = \frac{\Delta_2^2}{8} \cdot \frac{1}{\sigma^2} = \frac{1}{4}\, \frac{\Delta_2^2}{2\sigma^2} = \frac{\beta}{4},$$

and

$$\frac{C_0}{4}\, s^2 = \frac{C_0}{4} \cdot \frac{1}{\sigma^4} = \underbrace{\frac{C_0}{\Delta_2^4}}_{\triangleq C} \left( \frac{\Delta_2^2}{2\sigma^2} \right)^2 = C\, \beta^2.$$

Hence

$$I(X; e_g(X)) \ \geq \ \frac{\beta}{4} - C\, \beta^2.$$

**Conclusion.** Since $\beta \to 0$ implies $C\beta^2 = o(1)$, we have $I(e_h(X); e_g(X)) = I(X; e_g(X)) \geq \frac{\beta}{4} - o(\beta)$. Therefore the upper bound $I(e_h(X); e_g(X)) \leq \beta$ is tight up to a universal constant factor, establishing order-wise tightness. $\square$

### B.3 PROOF OF THEOREM 3

*Proof of Theorem 3 (Type I Error).* Let $n = |\mathcal{S}|$ and define

$$\widehat{I} \triangleq \widehat{I}_{\mathrm{KSG}}\big(e_{h_{\mathrm{ind}}}(X), e_g(X)\big), \qquad \mu_{\mathrm{ind}} \triangleq \mathbb{E}[\widehat{I}].$$

**Estimator.** We use the KSG estimator (Kraskov et al., 2004):

$$\widehat{I}_{\mathrm{KSG}} = \psi(k) - \frac{1}{n}\sum_{i=1}^{n}\Big[\psi\big(n_x(i)+1\big) + \psi\big(n_y(i)+1\big)\Big] + \psi(n),$$

where $k$ is fixed, $\psi$ is the digamma function, $n_x(i)$ is the number of samples (excluding $i$) whose $Z_1$-distance to sample $i$ does not exceed the joint $k$-NN radius $\varepsilon_i$, and $n_y(i)$ is defined analogously in $Z_2$ space. Assume embeddings are bounded: $\|Z_{1,i}\|_2 \leq B_1$, $\|Z_{2,i}\|_2 \leq B_2$, so all distances and counts are well-defined.

**Bounded differences.** Replace a single sample $(Z_{1,r}, Z_{2,r})$ by $(Z'_{1,r}, Z'_{2,r})$. This can affect: (i) the $r$-th summand itself, (ii) any sample $j \neq r$ for which $r$ falls inside the ball of radius $\varepsilon_j$ in $Z_1$ or $Z_2$ (thus changing $n_x(j)$ or $n_y(j)$ by at most 1). For $k$-nearest neighbour type estimators, each point can belong to at most $k$ such neighbour sets in each marginal space, so at most $2k$ other summands change. Hence, no more than $(2k+1)$ summands are affected. Since $\psi$ is monotone and for $m \in [1, n]$ we have $|\psi(m_1) - \psi(m_2)| \leq \log n$, each affected summand

$$\psi(n_x(i)+1) + \psi(n_y(i)+1)$$

changes by at most $2\log n$. Therefore the total change in the average is bounded by

$$\big|\widehat{I} - \widehat{I}'\big| \leq \frac{(2k+1)\cdot 2\log n}{n} = \frac{C_k}{n}, \qquad C_k \triangleq 2(2k+1)\log n.$$

Thus McDiarmid's inequality (McDiarmid et al., 1989) applies with $c_i = C_k/n$ for all $i$.

**Applying McDiarmid (upper tail).** For any $t > 0$,

$$\Pr\Big(\widehat{I} - \mu_{\mathrm{ind}} \geq t\Big) \leq \exp\Big(-\frac{2t^2}{\sum_{i=1}^{n}(c_i)^2}\Big) = \exp\Big(-\frac{2nt^2}{C_k^2}\Big).$$

Take $t = \tau(\sigma, Q) - \mu_{\mathrm{ind}}$. By design (independently trained $h_{\mathrm{ind}}$), $\mu_{\mathrm{ind}} < \tau(\sigma, Q)$, so $t > 0$. Therefore

$$\Pr\Big[\widehat{I} - \mu_{\mathrm{ind}} > \tau(\sigma, Q)\Big] \leq \exp\left(-\frac{2n\big(\tau(\sigma, Q) - \mu_{\mathrm{ind}}\big)^2}{C_k^2}\right).$$

In the ideal case $\mu_{\mathrm{ind}} \approx 0$ (true MI near zero and KSG is consistent), this becomes

$$\Pr\Big[\widehat{I} > \tau(\sigma, Q)\Big] \leq \exp\left(-\frac{2n\,\tau(\sigma, Q)^2}{\big[2(2k+1)\log n\big]^2}\right) \triangleq \gamma_1,$$

which is the stated Type I bound. $\qquad\qquad\square$

*Proof of Theorem 3 (Type II Error).* Let $n = |\mathcal{S}|$ and define

$$\widehat{I} \triangleq \widehat{I}_{\mathrm{KSG}}\big(e_{h_{\mathrm{sur}}}(X), e_g(X)\big), \qquad \mu_{\mathrm{sur}} \triangleq \mathbb{E}[\widehat{I}].$$

We again use the KSG estimator with the same bounded embedding and i.i.d. sample assumptions as in the Type I proof.

**Bounded differences.** The identical argument shows that replacing a single sample changes $\widehat{I}$ by at most $C_k/n$ with

$$C_k = 2(2k+1)\log n.$$

**Applying McDiarmid (lower tail).** We want

$$\Pr\Big[\widehat{I} \leq \tau(\sigma, Q)\Big] = \Pr\Big[\mu_{\mathrm{sur}} - \widehat{I} \geq \mu_{\mathrm{sur}} - \tau(\sigma, Q)\Big].$$

Assume the surrogate is sufficiently close to $g$ so that $\mu_{\text{sur}} > \tau(\sigma, Q)$; set $t = \mu_{\text{sur}} - \tau(\sigma, Q) > 0$. McDiarmid's inequality yields

$$\Pr\left[\widehat{I} \leq \tau(\sigma, Q)\right] \leq \exp\left(-\frac{2n\,t^2}{C_k^2}\right) = \exp\left(-\frac{2n\big(\mu_{\text{sur}} - \tau(\sigma, Q)\big)^2}{\big[2(2k+1)\log n\big]^2}\right) \triangleq \gamma_2.$$

Because $\tau(\sigma, Q) \to \beta(\sigma, \delta)(1 - \rho)$ as $n \to \infty$ and $\mu_{\text{sur}}$ stays above this limit by a positive margin, the exponent diverges to $-\infty$, hence $\gamma_2 \to 0$. $\qquad\square$

## C  REPRODUCIBILITY

### C.1  DATASET STATISTICS

This section provides an overview of the datasets used across different modalities. For the image classification tasks, we employ most commonly used CIFAR-10 and CIFAR-100, with detailed statistics summarized in Table 4.

Table 4: Statistics of commonly used real-world image classification datasets.

| Dataset | Classes | #Images | Train / Test Split | Resolution |
|---|---|---|---|---|
| CIFAR-10 | 10 | 60,000 | 50,000 / 10,000 | $32 \times 32$ RGB |
| CIFAR-100 | 100 | 60,000 | 50,000 / 10,000 | $32 \times 32$ RGB |

For the graph classification tasks, we employ most commonly used ENZYMES and PROTEINS datasets, with their corresponding statistics reported in Table 5.

Table 5: Statistics of commonly used real-world graph classification datasets.

| Dataset | #Graphs | Avg. #Nodes | Avg. #Edges | #Features | #Classes |
|---|---|---|---|---|---|
| ENZYMES | 600 | $\sim$32.6 | $\sim$124.3 | 3 | 6 |
| PROTEINS | 1,113 | $\sim$39.1 | $\sim$145.6 | 3 | 2 |

**Dataset Partitioning.** For dataset partitioning, we strictly follow the default train–test split. In addition, we introduce two auxiliary subsets: the query set and the verification set. Specifically, the query set is defined as a randomly sampled subset of the training set, with its size controlled by the query budget Q. The verification set is defined as a randomly sampled subset of the testing set, with its size determined according to practical requirements. Importantly, we ensure that the query set and verification set are strictly non-overlapping.

### C.2  BACKBONE MODEL IMPLEMENTATIONS

**Image Classification.** For the image classification tasks, we adopted four widely used convolutional neural network backbones: ResNet-50 (He et al., 2016), VGG-16 (Simonyan & Zisserman, 2014), DenseNet (Huang et al., 2017), and GoogLeNet (Szegedy et al., 2015). All of these models were directly loaded from the `torchvision.models` module to ensure standardized implementations and consistency across experiments. Using the official implementations avoids discrepancies in architecture design, parameter initialization, and optimization setups, thereby guaranteeing comparability of results.

**Graph Classification.** For the graph classification tasks, we employed four representative graph neural network backbones: GCN (Kipf & Welling, 2016), GAT (Veličković et al., 2017), Graph-SAGE (Hamilton et al., 2017), and SSGC (Zhu & Koniusz, 2021). These models were directly taken from the `torch_geometric.nn` package. Leveraging the established implementations provided by PyTorch Geometric ensures both correctness and reproducibility, while also aligning with commonly adopted practice in the literature.

Overall, these backbone choices span widely used CNN and GNN architectures, which ensures both the broad applicability of our evaluation and the fairness of comparison across different defense methods.

## C.3 BASELINE IMPLEMENTATIONS

For the **image classification** tasks, we selected four representative baselines: Backdooring (Adi et al., 2018), EWE (Jia et al., 2021), IPGuard (Cao et al., 2021), and UAP (Peng et al., 2022). Since the official implementations were not publicly available, we carefully re-implemented all methods by strictly following the descriptions provided in their original papers and reproduced the training/inference pipelines to the best of our ability. The details are as follows:

- **Backdooring**: Following the paper, we randomly generated watermark keys and assigned them with random labels.
- **EWE**: We implemented the soft nearest neighbor loss (SNNL) as described in the original work and deployed it during the training process.
- **IPGuard**: We generated fingerprint points according to the original design and assigned random labels for watermarking.
- **UAP**: We followed the original formulation by generating universal adversarial perturbations and conducting inference on the fingerprint points.

For the **graph classification** tasks, we selected four representative baselines: RandomWM (Zhao et al., 2021), BackdoorWM (Xu et al., 2023), SurviveWM (Wang et al., 2023), ImperceptibleWM (Zhang et al., 2024b). Due to the same limitation of unavailable official code, we also reproduced the baselines faithfully based on the textual descriptions:

- **RandomWM**: Randomly sampled watermark points and assigned random labels.
- **BackdoorWM**: Randomly sampled watermark points, modified their node features, and assigned them with a fixed label.
- **SurviveWM**: Randomly sampled watermark points, assigned random labels, and trained the model with the soft nearest neighbor loss (SNNL) objective.
- **ImperceptibleWM**: Applied perturbations directly on input batches and trained the model on the perturbed data.

Overall, although no official source codes were available, our implementations strictly adhered to the methodological descriptions provided in the original works, ensuring that the reproduced baselines are as faithful and comparable as possible.

## C.4 MODEL EXTRACTION ATTACK SETTINGS

Since our proposed method is a certified defense against model extraction attacks (MEAs), it is essential to evaluate its effectiveness under practical attack scenarios. To ensure generality across all baseline defenses, we adopt two representative and widely studied MEA strategies. First, we consider knowledge distillation (Romero et al., 2014), the most commonly used paradigm for model extraction, where an adversary trains a surrogate model by minimizing the divergence between the surrogate predictions and the outputs of the protected target model. This captures the practical threat that a black-box adversary can replicate the functionality of the target model through systematic querying. In addition, we follow the Knockoff (Orekondy et al., 2019a) framework and evaluate random querying, where the adversary issues queries sampled from an unlabeled data distribution without task-specific optimization. This setting reflects a weaker but still realistic adversary that relies purely on large-scale queries. Together, these two attack strategies cover both structured and unstructured extraction scenarios, providing a balanced evaluation of our certified defense under practical MEA conditions.

## C.5 PRACTICAL DEPLOYMENT OF CREDIT

In this subsection, we provide a detailed analysis of how CREDIT can be deployed in practical production settings.

**Computing $\tau$.** To begin, we must determine known parameters such as the embedding dimension d (set to 1024 in our implementation) and the size of the verification set $\mathcal{V}$ (set to 1000). We also specify the attacker's query budget $Q$, which reflects the standard used for ownership verification.

For example, if we set $Q = 5000$, then we can claim that any surrogate model trained with at least 5000 queries will be verifiable under our scheme, with the corresponding threshold $\tau$ yielding the associated maximum Independent False Alarm (Type I error) probability $\gamma_1$ and Surrogate Missed Detection (Type II error) probability $\gamma_2$.

**Computing $\sigma^\star$.** As discussed earlier, the optimal $\sigma$ is obtained by performing a grid search over a set of reasonable candidate values. For each $\sigma$, we compute $\gamma_1$ and $\gamma_2$, along with the utility entropy and verification entropy. By optimizing the objective function, we then can obtain the optimal $\sigma^\star$.

## C.6 HARDWARE INFORMATION

All training, inference, and efficiency evaluations were carried out on a high-performance computing server equipped with an NVIDIA RTX 6000 Ada GPU. The system is powered by an AMD EPYC 7763 processor with 64 cores (128 threads) operating at 2.45 GHz. The server is provisioned with 1 TB of Samsung DDR4 registered and buffered memory running at 3200 MT/s, ensuring sufficient computational and memory resources for large-scale experiments.

# D SUPPLEMENTAL EXPERIMENTS

## D.1 ROBUSTNESS OF THE KSG ESTIMATOR

To examine whether the choice of nearest neighbors $k$ influences the mutual information (MI) used in CREDIT, we tested four values, namely $k \in \{3, 5, 7, 10\}$, and computed the mutual information between the target model and the independent model under each setting. Each value was evaluated over repeated trials and we report the mean and standard deviation.

| $k$ | MI (mean $\pm$ std) |
|---|---|
| 3 | $0.0492 \pm 0.0028$ |
| 5 | $0.0422 \pm 0.0021$ |
| 7 | $0.0402 \pm 0.0026$ |
| 10 | $0.0470 \pm 0.0025$ |

These results show that the KSG estimator is stable across different values of $k$ and CREDIT remains consistent without requiring fine tuning of this hyperparameter.

## D.2 ROBUSTNESS AGAINST QUERY-AVERAGING AND DENOISING ATTACKS

A potential concern for Gaussian-mechanism–based defenses is that an adversary may attempt to reduce the variance of injected noise by issuing repeated queries for the same input and averaging the responses. Averaging $m$ independent Gaussian samples reduces variance by a factor of $1/m$, which raises the question of whether such a strategy might increase the mutual information (MI) available to the adversary and thereby weaken the verification guarantees of CREDIT.

To evaluate this scenario, we consider a fixed total query budget $q = 0.1$, chosen sufficiently large for a standard model extraction attack to train a functional surrogate model. We then vary the repetition factor $m$, which reduces the number of *distinct* queries to $q/m$. Since model extraction fundamentally relies on the diversity of query–response pairs, a reduction in distinct queries directly restricts the exploitable information available to the adversary.

Table 6: Effect of repeated queries under a fixed total query budget $q$. Increasing $m$ reduces the number of distinct queries and significantly degrades the utility of the extracted surrogate.

| Repeat $m$ | Distinct Query Ratio | Surrogate Test Acc |
|---|---|---|
| 1 | 0.1000 | 0.7303 |
| 2 | 0.0500 | 0.5125 |
| 4 | 0.0250 | 0.2833 |
| 8 | 0.0125 | 0.1997 |
| 10 | 0.0100 | 0.1720 |

As the Table 6 shows, increasing $m$ leads to a rapid collapse in surrogate model accuracy. Although averaging reduces the variance of Gaussian noise, it does not provide additional information about the decision boundary because the number of distinct queries shrinks proportionally to $q/m$. Thus, repeated queries consume the attacker's budget without meaningfully improving the fidelity of the surrogate model. These results demonstrate that query-averaging attacks are inherently inefficient and economically impractical. The loss of unique queries outweighs the marginal benefit of noise averaging, thereby preserving the robustness of CREDIT even without additional defenses.

### D.3    COMPARISON WITH MODEL INTEGRITY VERIFICATION METHODS

A recent model integrity work (He et al., 2024) has received attention and proposes the MiSentry approach for verifying whether a deployed model has been tampered with. To ensure comprehensiveness in our baseline comparison, we additionally include MiSentry in our evaluation and examine whether it can be adapted to the model extraction setting studied in this paper.

To assess whether MiSentry can be adapted to the model extraction setting, we implemented a faithful adaptation of MiSentry and applied it to the surrogate models extracted under our experimental protocol. We then evaluated ownership verification performance using the standard AUC metric.

Table 7: Ownership verification AUC under the model extraction setting.

| Method | ResNet | VGG | DenseNet | GoogLeNet |
|---|---|---|---|---|
| Backdooring | 58.64 | 51.85 | 74.07 | 80.25 |
| EWE | 51.24 | 62.96 | 48.77 | 62.35 |
| IPGuard | 40.74 | 61.11 | 67.90 | 50.62 |
| UAP | 52.47 | 67.28 | 72.22 | 79.63 |
| MiSentry | 47.30 | 54.88 | 59.20 | 61.44 |
| CREDIT (ours) | **100.0** | **100.0** | **100.0** | **100.0** |

The results in Table 7 show that MiSentry, even when adapted to our setting, exhibits performance comparable to other watermarking and fingerprinting methods and fails to achieve reliable ownership verification after model extraction. This confirms that integrity-based fingerprints do not survive the extraction process. By contrast, CREDIT achieves perfect identification across all tested architectures, highlighting both its robustness and the fundamental difference between our setting and prior integrity-oriented work.

### D.4    CERTIFIED OWNERSHIP VERIFICATION UNDER BOUNDED ADVERSARIAL UTILITY

To examine the robustness of CREDIT under worst-case adversarial manipulations, we construct a family of *worst-case decorrelation attacks* designed to maximally disrupt the verification signal. In this setting, the adversary is given a bounded utility budget $\delta \in \{0\%, 3\%, 5\%, 10\%\}$, which specifies the maximum allowable degradation of the surrogate model's clean accuracy. Within this constraint, the adversary is free to apply decorrelation operations that minimize the statistical dependence between the surrogate model and the target model while still respecting the required utility bound. This setting represents an intentionally challenging evaluation regime tailored to test the limits of our certified ownership verification framework.

Table 8: Worst-case decorrelation attacks under bounded utility loss.

| Attack Type | Utility Budget $\delta$ | Test Acc | MI | Verification AUC |
|---|---|---|---|---|
| None | 0% | 0.7322 | 2.2887 | 1.0000 |
| Decorrelation | 3% | 0.7301 | 2.2611 | 1.0000 |
| Decorrelation | 5% | 0.7182 | 2.2498 | 1.0000 |
| Decorrelation | 10% | 0.6947 | 2.2023 | 1.0000 |

As shown in Table 8, although stronger decorrelation consistently harms both mutual information and surrogate utility as the allowed budget increases, CREDIT maintains perfect ownership verification (AUC = 1.0000) in all scenarios. These results demonstrate that even under adversarial

manipulations explicitly optimized to challenge our verification signal, the verification guarantee offered by CREDIT remains robust.

### D.5 EXTENSION TO NLP MODALITY: WORD2VEC OWNERSHIP VERIFICATION

To further examine the modality generality of CREDIT, we additionally evaluate our framework in a natural language processing setting. Specifically, we train a Word2Vec (Mikolov et al., 2013) model on the STSb dataset (Cer et al., 2017) to obtain sentence-level embeddings, and then apply CREDIT to verify ownership of suspicious models extracted under our standard protocol. This setup provides a lightweight yet controlled environment for assessing whether our certified ownership verification approach extends beyond computer vision and graph modalities.

Table 9: Ownership verification results on a Word2Vec model trained on STSb.

| Model | MI | Verification AUC |
|---|---|---|
| Word2Vec | 1.2331 | 1.0000 |

The results in Table 9 show that CREDIT achieves perfect ownership verification (AUC = 1.0000) even in this NLP setting, with a clear separation of MI values between target and independent models. This demonstrates that the proposed certified ownership verification framework is broadly applicable across modalities, including computer vision, graph data, and natural language processing.

## E STATEMENTS

### E.1 LLM USAGE STATEMENT

Large Language Models (LLMs) were not used for generating research ideas, designing algorithms, or conducting experiments. LLMs were only used to assist in polishing the presentation of the manuscript, such as improving grammar, clarity, and flow of text. All technical content, including methods, proofs, and experiments, was fully developed and validated by the authors.

