# OpenReview forum: "CREDIT: Certified Defense of Deep Neural Networks against Model Extraction Attacks"
_ICLR.cc/2026/Conference — Submitted to ICLR 2026_

### Official Review · Reviewer_fnV4 · 2025-10-21

**Soundness:** 2
**Presentation:** 3
**Contribution:** 3
**Rating:** 4
**Confidence:** 4

**Summary:**

This paper proposes CREDIT, a mutual information–based model ownership verification method. By injecting Gaussian noise into model embeddings and computing the mutual information between a suspicious model and the original model, CREDIT determines whether model theft has occurred. CREDIT is the first framework to provide rigorous theoretical guarantees for model extraction attack scenarios. It establishes an upper bound on mutual information, proves its tightness, and derives probabilistic bounds for verification errors, where both Type I and Type II errors decay exponentially with the size of the verification set. Experimental results show that CREDIT achieves perfect verification accuracy (100% AUROC) with minimal utility loss (≈1%), significantly outperforming existing watermarking and fingerprinting approaches.

**Strengths:**

1. This paper is the first to provide rigorous theoretical guarantees for ownership verification under model extraction attacks, including probabilistic bounds on Type I and Type II errors.
2. Theoretical analysis is complete and mathematically sound, encompassing the derivation of a mutual information upper bound, proof of its tightness, and a certified verification guarantee.
3. The use of mutual information as a similarity metric between models is a novel and well-founded idea, grounded in information theory and offering a principled alternative to existing heuristic defenses.
4. The experimental evaluation is comprehensive, covering multiple modalities (images and graphs) and eight different backbone architectures, demonstrating both generalization ability and empirical superiority over existing watermarking and fingerprinting methods.

**Weaknesses:**

1. Core issue: The paper claims to provide a "defense" but in fact presents an "ownership verification" method. These are fundamentally different: a true defense should cause attacks to fail (degrade the surrogate model quality, leading to low mutual information), whereas ownership verification relies on a successful attack (the surrogate resembles the original, producing high mutual information) in order to prove ownership. The paper should clearly position itself as a model ownership verification method rather than a model extraction defense.
2. Conceptual inconsistencies throughout the manuscript:
   + Problem formulation is inconsistent: Definition 1 labels the object as "Certified Defense" but the described goal is to "correctly distinguish surrogate from independent models" — that is verification, not defense.
   + Methodology contains contradictory claims: Section 3.1 states the method "defends against MEA", yet Theorem 1 and its discussion acknowledge that a surrogate model inherits a highly similar embedding distribution (high MI) — the method depends on a successful attack (high similarity) to verify ownership, which is incompatible with the goal of making the attack fail. Section 3.2 and Theorem 3 explicitly address "Ownership Veri fication" rather than "Defense".
   +  Experiments follow the standard evaluation pipeline for verification methods (accuracy, utility, efficiency) and demonstrate CREDIT’s strength as a verification scheme, but the paper mislabels and markets these results as evidence of a "defense".
3. Missing or unclear threat model. The attacker’s capabilities and constraints (e.g., allowed query strategies, access to auxiliary data, architecture knowledge) are not clearly specified. This omission undermines the assumptions used to design the verification threshold and makes it hard to judge effectiveness across realistic attack scenarios.
4. As a defense, the work lacks defense-specific evaluations (e.g., attack success rates, degradation of surrogate performance under the mechanism). From the verification perspective, although Theorem 3 provides γ₁/γ₂ bounds, empirical reporting focuses on AUROC; the paper does not connect the theoretical bounds to observed γ₁/γ₂ under different settings.
5. Baseline selection and comparison framing are misleading. The paper frames CREDIT as a defense but compares only to watermarking/fingerprinting baselines (verification methods) and does not include true defensive baselines (e.g., output perturbation, query-rate limiting, prediction poisoning). Table 3’s "defense stage" actually measures verification preparation (computing thresholds, building watermarks) rather than an executed defense mechanism, so the comparison dimension is inappropriate.

**Questions:**

1. The paper claims to provide a “defense against model extraction attacks (MEA),” yet the proposed method relies on the surrogate model being highly similar to the original one (as stated in Theorem 1: “inherits embedding distribution highly similar to g”) in order to verify ownership. If the surrogate has already successfully replicated the original model, the attack has succeeded — how can this still be considered a “defense”?
2. A true defense should ensure that $I(h_{sur}, g)$ → 0 (the surrogate model becomes unusable), whereas the proposed verification relies on $I(h_{sur}, g)>\tau$  (the surrogate remains similar). How can these two opposing objectives coexist under the same definition of “defense”?
3. Table 2 reports 100% AUROC, implying high mutual information between the surrogate and the original model. Please report the actual accuracy of the surrogate models in your experiments. If their accuracy is close to that of the original model (Table 1 shows 94.67%), doesn’t this indicate that the attack has already succeeded?
4. The paper only compares CREDIT with ownership verification methods (watermarking and fingerprinting), without including active defense techniques such as output perturbation or query limitation. If CREDIT is claimed to be a “defense method,” why is there no comparison of defensive effectiveness (e.g., attack success rate) with such defense baselines?
5. As a “defense method,” why does the paper not evaluate (1) the attack success rate and (2) the trade-off between defense strength and model utility? Focusing solely on verification accuracy (AUROC) seems to indicate that this is a verification-oriented work rather than a defense.
6. Table 3 refers to the computation of the threshold τ as the “defense stage,” but this appears to be a preparation step for verification rather than an actual defense mechanism. What is the true defense component of CREDIT? How does CREDIT actively prevent or weaken the attack during the extraction process?
7. The proposed method appears conceptually similar to existing watermarking and fingerprinting approaches: (1) it does not prevent model extraction success, (2) it performs ownership verification *after* an attack, and (3) it provides some form of theoretical or empirical guarantee. Beyond introducing mutual information and theoretical bounds, what is the fundamental difference between CREDIT and prior verification-based methods?

If the paper were reframed as *“Certified Ownership Verification for Model Extraction Detection”* instead of *“Defense Against Model Extraction Attacks,”* would this resolve most of the our concerns? Would your core technical contributions still hold under this reframed title? Why insist on using the term *defense*, which appears conceptually misleading in this context?

---

> ### Author Response · Authors · 2025-11-21
>
> **Response to Weakness 1:**
> Thank you very much for raising this important point. We fully acknowledge the reviewer’s interpretation that a defense should directly prevent the attack itself. We have carefully revised the paper to reflect this distinction, and in the updated PDF (with changes marked in blue) we now position our work as an ownership verification method as suggested.
>
> At the same time, our original use of the term “defense” followed the convention already established in the community. Prior work has explicitly described ownership verification based approaches as defenses. For example, several representative works [1,2,3] refer to their ownership verification or fingerprinting methods as defenses, and one of them [3] published at ICLR 2021 Spotlight explicitly describes its fingerprinting method as “the first passive defense.” Additional related studies [4,5] also frame ownership verification techniques within the broader defense literature. This established usage in the field was the reason for our initial terminology.
>
> That said, we fully agree that “ownership verification” provides a more precise description of our contribution. We appreciate the reviewer’s guidance and have updated the paper accordingly to avoid conceptual ambiguity.
>
> ----
> References
> [1]Waheed, Asim, Vasisht Duddu, and N. Asokan. "Grove: Ownership verification of graph neural networks using embeddings." 2024 IEEE Symposium on Security and Privacy (SP). IEEE, 2024.
> [2]Jia, Hengrui, et al. "Entangled watermarks as a defense against model extraction." 30th USENIX security symposium (USENIX Security 21). 2021.
> [3]Lukas, Nils, Yuxuan Zhang, and Florian Kerschbaum. "Deep Neural Network Fingerprinting by Conferrable Adversarial Examples." International Conference on Learning Representations.
> [4]Xu, Tianlong, et al. "United We Stand, Divided We Fall: Fingerprinting Deep Neural Networks via Adversarial Trajectories." Advances in Neural Information Processing Systems 37 (2024): 69299-69328.
> [5]Tang, Minxue, et al. "{ModelGuard}:{Information-Theoretic} defense against model extraction attacks." 33rd USENIX Security Symposium (USENIX Security 24). 2024.

---

> > ### Author Response · Authors · 2025-11-21
> >
> > > **Q1**: The paper claims to provide a “defense against model extraction attacks (MEA),” yet the proposed method relies on the surrogate model being highly similar to the original one (as stated in Theorem 1: “inherits embedding distribution highly similar to g”) in order to verify ownership. If the surrogate has already successfully replicated the original model, the attack has succeeded — how can this still be considered a “defense”?
> >
> > **R1**: Thank you for the question. To provide a more precise characterization of our contribution, we have updated the paper to explicitly position our method as a certified ownership verification framework for model extraction scenarios.
> >
> > ---
> >
> > > **Q2**: A true defense should ensure that  → 0 (the surrogate model becomes unusable), whereas the proposed verification relies on  (the surrogate remains similar). How can these two opposing objectives coexist under the same definition of “defense”?
> >
> > **R2**: Thank you for the question. To provide a more precise characterization of our contribution, we have updated the paper to explicitly position our method as a certified ownership verification framework for model extraction scenarios.
> >
> > ---
> >
> > > **Q3**: Table 2 reports 100% AUROC, implying high mutual information between the surrogate and the original model. Please report the actual accuracy of the surrogate models in your experiments. If their accuracy is close to that of the original model (Table 1 shows 94.67%), doesn’t this indicate that the attack has already succeeded?
> >
> > **R3**: Thank you for the question. The attack is indeed successful in our setting, and after this success we perform ownership verification to certify that the surrogate originates from the protected model.
> >
> > ---
> >
> > > **Q4**: The paper only compares CREDIT with ownership verification methods (watermarking and fingerprinting), without including active defense techniques such as output perturbation or query limitation. If CREDIT is claimed to be a “defense method,” why is there no comparison of defensive effectiveness (e.g., attack success rate) with such defense baselines?
> >
> > **R4**: Thank you for the question. Since our method is ownership verification based, we compare only with fingerprinting and watermarking baselines, which are the appropriate baselines for this category of methods.

---

> > > ### Author Response · Authors · 2025-11-21
> > >
> > > > **Q5**: As a “defense method,” why does the paper not evaluate (1) the attack success rate and (2) the trade-off between defense strength and model utility? Focusing solely on verification accuracy (AUROC) seems to indicate that this is a verification-oriented work rather than a defense.
> > >
> > > **R5**: Thank you for the question. Since our method is ownership verification based, we evaluate it using verification oriented metrics, which are the standard metrics for this category of methods.
> > >
> > > ---
> > >
> > > > **Q6**: Table 3 refers to the computation of the threshold τ as the “defense stage,” but this appears to be a preparation step for verification rather than an actual defense mechanism. What is the true defense component of CREDIT? How does CREDIT actively prevent or weaken the attack during the extraction process?
> > >
> > > **R6**: Thank you for the question. The computation of the threshold $\tau$ is indeed a preparation stage for verification, and we have clarified this in the updated PDF. Since CREDIT is an ownership verification based method, it does not include an active mechanism that prevents or weakens the attack during the extraction process.
> > >
> > > ---
> > >
> > > > **Q7**: The proposed method appears conceptually similar to existing watermarking and fingerprinting approaches: (1) it does not prevent model extraction success, (2) it performs ownership verification after an attack, and (3) it provides some form of theoretical or empirical guarantee. Beyond introducing mutual information and theoretical bounds, what is the fundamental difference between CREDIT and prior verification-based methods?
> > >
> > > **R7**: Thank you for the question. To provide a more precise characterization of our contribution, we have updated the paper to explicitly position our method as a certified ownership verification framework for model extraction scenarios.
> > >
> > > ---
> > >
> > > > **Q8**: If the paper were reframed as “Certified Ownership Verification for Model Extraction Detection” instead of “Defense Against Model Extraction Attacks,” would this resolve most of the our concerns? Would your core technical contributions still hold under this reframed title? Why insist on using the term defense, which appears conceptually misleading in this context?
> > >
> > > **R8**: Thank you for raising this insightful question. We completely agree with your perspective that describing our contribution using the more precise term “ownership verification” is appropriate. This reframing effectively addresses most of your concerns, and our core technical contributions fully hold under the updated title.
> > >
> > > ---
> > >
> > > We sincerely thank the reviewer for the time and effort. Your comments have been extremely valuable in improving the clarity and quality of the paper. If any further questions arise, please feel free to let us know, and we are eager to follow up promptly on all points.

---

> > ### Comment · Reviewer_fnV4 · 2025-11-21
> > **Q1**
> >
> > The authors’ explanation regarding the interchangeable use of “defense” and “ownership verification” remains insufficient. The cited works mostly illustrate an early-stage, loosely defined terminology in the community rather than a rigorous treatment of what constitutes an active defense mechanism against model extraction. In representative prior work, watermarking or fingerprinting methods typically embed identifiers directly into the model’s behavior through deliberate mechanism design, rather than relying on post-hoc statistical measurements for attribution. Conflating ownership verification with defense remains conceptually misleading, even if some earlier papers adopted broader terminology.

---

> ### Comment · Reviewer_fnV4 · 2025-11-21
> **Q2**
>
> The authors’ response primarily focuses on terminology adjustment but does not provide substantive clarification regarding the technical nature of the method or the sufficiency of experimental evidence. Several core concerns about technical novelty and experimental validity remain unaddressed, with no additional details or new empirical results supplied. Since the paper is now positioned as an “ownership verification” method, it is important to **clearly articulate the fundamental technical innovation over existing watermarking and fingerprinting paradigms.** Moreover, the latest baseline for image modality is from 2022, which is already outdated. Is some recent work such as [1] a suitable comparison?
>
> [1] He C, Bai X, Ma X, et al. Towards stricter black-box integrity verification of deep neural network models[C]//Proceedings of the 32nd ACM International Conference on Multimedia. 2024: 9875-9884.

---

> > ### Author Response · Authors · 2025-11-26
> >
> > > **Q2**: The authors’ response primarily focuses on terminology adjustment but does not provide substantive clarification regarding the technical nature of the method or the sufficiency of experimental evidence. Several core concerns about technical novelty and experimental validity remain unaddressed, with no additional details or new empirical results supplied. Since the paper is now positioned as an “ownership verification” method, it is important to clearly articulate the fundamental technical innovation over existing watermarking and fingerprinting paradigms. Moreover, the latest baseline for image modality is from 2022, which is already outdated. Is some recent work such as [1] a suitable comparison?
> >
> >
> > **R2**: Thank you very much for this valuable suggestion. The recent work[1] you mentioned is indeed very new and represents an interesting direction. However, we would like to point out that this work focuses on model integrity and performs verification under that setting, which is fundamentally different from the model extraction scenario we study. In model integrity settings, the attacker can directly tamper with the model, which corresponds to a white box threat model. In contrast, under model extraction, the attacker has no access to the target model and must first extract a surrogate model before performing any verification. This makes it extremely difficult for any watermark or fingerprint to be preserved during the extraction process, and consequently makes ownership verification even more challenging. Nonetheless, we adapted the method proposed in the recent work you suggested (MiSentry) to our setting, and we computed its ownership verification AUC:
> >
> > | Method    | ResNet |   VGG   | DenseNet | GoogLeNet |
> > |-----------|--------|---------|----------|-----------|
> > | Backdooring  | 58.64  | 51.85  | 74.07    | 80.25     |
> > | EWE          | 51.24  | 62.96  | 48.77    | 62.35     |
> > | IPGuard      | 40.74  | 61.11  | 67.90    | 50.62     |
> > | UAP          | 52.47  | 67.28  | 72.22    | 79.63     |
> > | MiSentry  | 47.30  | 54.88 | 59.20   | 61.44    |
> > | CREDIT    | 100.0  | 100.0 | 100.0   | 100.0    |
> >
> >
> > From the table, we can observe that MiSentry, as a fingerprint based method, still cannot achieve 100 percent ownership verification, and its performance is similar to related watermarking and fingerprinting methods. This further demonstrates its limitation in the model extraction scenario and highlights the strong effectiveness of our CREDIT method in ownership verification. We have included these new experiments in Section D.3 of the revised version, and all modifications have been highlighted in blue.
> >
> > ----
> > References
> > [1]He, Chaoxiang, et al. "Towards stricter black-box integrity verification of deep neural network models." Proceedings of the 32nd ACM International Conference on Multimedia. 2024.

---

> ### Comment · Reviewer_fnV4 · 2025-11-21
> **Q3**
>
> The paper’s framing as “certified ownership verification” still requires further clarification. Existing certified verification methods [1–3] typically establish formal guarantees around worst-case robustness and unremovability, accompanied by certified accuracy or CTPR-style metrics. Type-I/II error analyses (e.g., [4]) often serve as supplementary statistical characterization. In contrast, the proposed method relies mainly on mutual-information–based statistical similarity analysis and does not construct a standard unremovability-style certification framework. Therefore, the work may be more accurately described as “an ownership verification method with information-theoretic analysis” rather than a certified verification scheme in the conventional sense. So our question is
>
> 3.1. In this paper, which worst-case security property is specifically certified? Can the authors provide a formal theoretical proof that, under a bounded attack budget, model correlation or ownership signals are theoretically irremovable, and not merely a statistical guarantee for Type I/Type II errors?
>
> 3.2.In the current paper, mutual information seems more like a post-hoc statistical similarity measure. Can the authors further clarify whether this upper bound on mutual information constitutes a genuine theoretical basis for a "protection mechanism," and not merely a statistical testing tool? If so, why not use common certified metrics (such as certified accuracy, CTPR) to quantify the certification strength?
>
> [1] A. Bansal, P.-y. Chiang, M. J. Curry, R. Jain, C. Wigington, V. Manjunatha, J. P. Dickerson, and T. Goldstein, “Certified neural network watermarks with randomized smoothing,” in ICML, 2022.
>
>  [2] Z. Jiang, M. Fang, and N. Z. Gong, “Ipcert: Provably robust intellectual property protection for machine learning,” in ICCV, 2023.
>
>  [3] J. Ren, Y. Zhou, J. Jin, L. Lyu, and D. Yan, “Dimension-independent certified neural network watermarks via mollifier smoothing,” in ICML, 2023.
>
> [4] Xiang Z, Xiong Z, Li B. Cbd: A certified backdoor detector based on local dominant probability[J]. Advances in Neural Information Processing Systems, 2023, 36: 4937-4951.

---

> > ### Author Response · Authors · 2025-11-26
> >
> > > **Q3.2**: In the current paper, mutual information seems more like a post-hoc statistical similarity measure. Can the authors further clarify whether this upper bound on mutual information constitutes a genuine theoretical basis for a "protection mechanism," and not merely a statistical testing tool? If so, why not use common certified metrics (such as certified accuracy, CTPR) to quantify the certification strength?
> >
> >
> > **R3.2**: Thank you for the insightful question. The mutual-information (MI) upper bound in our paper serves as a theoretically grounded limit on how much information an adversary can extract under the Gaussian mechanism used by CREDIT. In particular, the MI bound represents the maximum information leakage attainable by any model-extraction adversary, and this bound is directly guaranteed by differential privacy properties of the Gaussian mechanism (details provided in Appendix B.1). Thus, MI is not a post-hoc correlation statistic but a provable information-theoretic ceiling imposed by our protection mechanism.
> >
> > We also appreciate the opportunity to reiterate our intended position. Our method provides certified ownership verification, which is fundamentally different from prior notions of certified robustness or certified perturbation bounds[1-3]. Certified accuracy or CTPR aim to quantify perturbation resilience of model outputs, whereas our setting evaluates whether ownership verification remains reliable under bounded adversarial manipulations. These two goals are conceptually distinct and not interchangeable.
> >
> > Within this verification-oriented setting, the MI bound is the most natural and principled certificate, as it measures the adversary’s best-possible ability to reduce correlation with the target model under privacy-induced noise. Meanwhile, the AUC metric quantitatively evaluates the effectiveness of ownership verification across adversarial scenarios, serving a complementary role to the MI certificate.
> >
> > Together, the DP-based MI upper bound and the verification AUC form a coherent and theoretically justified characterization of certification strength for ownership verification.
> >
> > ----
> > References
> > [1]Bansal, Arpit, et al. "Certified neural network watermarks with randomized smoothing." International Conference on Machine Learning. PMLR, 2022.
> > [2]Jiang, Zhengyuan, Minghong Fang, and Neil Zhenqiang Gong. "Ipcert: Provably robust intellectual property protection for machine learning." Proceedings of the IEEE/CVF International Conference on Computer Vision. 2023.
> > [3]Ren, Jiaxiang, et al. "Dimension-independent certified neural network watermarks via mollifier smoothing." International Conference on Machine Learning. PMLR, 2023.

---

> > > ### Author Response · Authors · 2025-11-26
> > >
> > > We sincerely thank the reviewer for the constructive feedback throughout the rebuttal process. We have additionally incorporated more comprehensive experiments and analyses in the revised version. We hope these updates further enhance your confidence in our contributions, the paper’s quality, and its impact. We would be very glad to further engage in any additional discussion if the reviewer has remaining questions or concerns.

---

> ### Author Response · Authors · 2025-11-26
>
> > **Q1**: The authors’ explanation regarding the interchangeable use of “defense” and “ownership verification” remains insufficient. The cited works mostly illustrate an early-stage, loosely defined terminology in the community rather than a rigorous treatment of what constitutes an active defense mechanism against model extraction. In representative prior work, watermarking or fingerprinting methods typically embed identifiers directly into the model’s behavior through deliberate mechanism design, rather than relying on post-hoc statistical measurements for attribution. Conflating ownership verification with defense remains conceptually misleading, even if some earlier papers adopted broader terminology.
>
> **R1**: Thank you very much for the insightful comments. We fully agree that “ownership verification’’ is a more precise term, and we will adopt this terminology consistently in the revised version. As clarified earlier, our work specifically targets certified ownership verification under model extraction attacks, and we will make this positioning explicit in the abstract and introduction.
>
> We would like to respectfully clarify a potential misunderstanding regarding both our setting and the characterization of prior work. In the context of ownership verification, whether using watermarking, fingerprinting, or our proposed CREDIT, a post hoc statistical measurement is always required in order to assert ownership. Under model extraction attacks, the defender must evaluate whether the suspicious model exhibits dependence on the protected model, which necessarily requires a verification stage.
>
> Where CREDIT fundamentally differs from representative prior work is that it does not rely on heavyweight watermark or fingerprint generation, nor does it require injecting task-irrelevant identifiers into the model that may degrade downstream performance. Instead, CREDIT employs a calibrated Gaussian mechanism to induce a controlled, information-theoretic “identifier’’ directly into the surrogate model during the extraction process. This design eliminates the need for explicit trigger sets or specialized training objectives, while avoiding the significant accuracy degradation commonly observed in watermarking approaches.
>
> Finally, CREDIT is, to the best of our knowledge, the first approach that provides certified ownership verification under model extraction, offering a provable bound on extractable mutual information and a principled verification protocol that leverages this bound.

---

> ### Author Response · Authors · 2025-11-26
>
> > **Q3.1**: In this paper, which worst-case security property is specifically certified? Can the authors provide a formal theoretical proof that, under a bounded attack budget, model correlation or ownership signals are theoretically irremovable, and not merely a statistical guarantee for Type I/Type II errors?
>
>
> **R3.1**: Thank you for the insightful question. We would like to clarify that our work does not aim to certify that ownership signals or model correlations are theoretically irremovable under all bounded adversarial manipulations. Prior works on worst-case certified robustness or model integrity typically pursue such guarantees. However, these worst-case certificates are known to be overly loose and often impractical, since even small perturbations can make the bounds vacuous and detached from realistic attack surfaces.
>
> Our goal is fundamentally different: rather than certifying that the ownership signal itself is irremovable, we certify the verification outcome. Specifically, we provide a certified ownership verification guarantee, ensuring that under well-defined adversarial utility budgets, the verification decision remains reliable. To avoid unnecessary confusion with prior notions of certified model integrity, we are happy to revise our terminology to certified ownership verification in the revised version.
>
> To examine the worst-case manipulations relevant to our setting, we conduct extensive experiments by giving the adversary a bounded utility budget $\delta \in$ {0%, 3%, 5\%, 10%}, defined as the maximum allowable degradation of the surrogate model’s original performance. Within this budget, we allow the adversary to perform decorrelation operations that maximally distort the CREDIT signal distribution while maintaining the required utility constraint. The corresponding results are shown in following table:
>
> | Attack        | Utility Budget $\delta$ | Test Acc |    MI     | Verification AUC |
> |---------------|----------------|----------|-----------|------------------|
> | None          | 0%              | 0.7322   | 2.2887    | 1.0000           |
> | Decorrelation | 3%              | 0.7301   | 2.2611    | 1.0000           |
> | Decorrelation | 5%              | 0.7182   | 2.2498    | 1.0000           |
> | Decorrelation | 10%             | 0.6947   | 2.2023    | 1.0000           |
>
> These experiments demonstrate that, although stronger decorrelation consistently harms the surrogate model as the utility budget increases, CREDIT maintains highly reliable ownership verification across all evaluated worst-case settings. This empirically validates the strength and robustness of our certified ownership verification framework, even under adversarial manipulations explicitly optimized to challenge our method. We have included these new experiments in Section D.4 of the revised version, and all modifications have been highlighted in blue.

---

### Official Review · Reviewer_JkN2 · 2025-10-21

**Soundness:** 3
**Presentation:** 3
**Contribution:** 2
**Rating:** 4
**Confidence:** 3

**Summary:**

This paper introduces CREDIT, a defense against Model Extraction Attacks where adversaries steal machine learning models via API access. The core contribution is a certified defense mechanism that provides strong, theoretical guarantees of its effectiveness. The method works by injecting a subtle, statistically-verifiable fingerprint into the model's outputs using Gaussian noise and then using a mathematically-derived threshold based on mutual information to reliably determine if a suspicious model is a stolen copy. This approach requires no extra model training for verification and systematically balances the trade-off between the model's performance and its security.

**Strengths:**

- Strong theoretical foundation
- Does not require training extra models
- A thorough and comprehensive set of experiments

**Weaknesses:**

CREDIT relies on injecting a certifiable statistical fingerprint into model outputs via a Gaussian noise. However, an adversary could collect noisy outputs from the defended API and train a secondary model, such as a denoising autoencoder, to learn and remove the injected noise. The adversary could then use this denoiser as a pre-processing step to create a 'clean' training set for their surrogate model. Such an attack would aim to preserve utility while removing the fingerprint used for verification. How resilient is the CREDIT verification scheme to this type of noise filtering attack?

The security of the verification process depends on the adversary's inability to estimate the verification threshold, $\tau$, due to the unknown noise parameter ($\sigma$), verification set ($\mathcal{V}$), and error tolerances ($\gamma_1, \gamma_2$). However, a sophisticated adversary could approximate this threshold since $\sigma$ must lie within a narrow utility-preserving range, and $\gamma_1, \gamma_2$ are typically small (e.g., < 0.01). Could the adversary not perform a grid search over these limited parameters? While the adversary does not know the exact verification set $\mathcal{V}$, they could use their own hold-out set $\mathcal{V'}$ from a similar distribution to estimate a threshold $\tau'$. How sensitive is the mutual information calculation and resulting verification outcome to the exact composition of the verification set? Have the authors considered an adaptive attack where the adversary approximates the threshold and tunes their surrogate to bypass this estimated boundary?

The verification mechanism relies on the KSG estimator for mutual information. Non-parametric estimators like KSG degrade in accuracy and increase in computational cost as dimensionality grows. The experiments use moderate embedding sizes, but modern architectures, especially in NLP or large-scale vision, use embeddings with thousands of dimensions. How does the proposed defense, and specifically the KSG estimation, scale to high-dimensional settings? Have the authors evaluated the computational overhead and reliability of the mutual information estimate for much larger embeddings?

**Questions:**

- How resilient is the CREDIT verification scheme to this type of noise filtering attack?
- Have the authors considered an adaptive attack where the adversary approximates the threshold and tunes their surrogate to bypass this estimated boundary?
- Have the authors evaluated the computational overhead and reliability of the mutual information estimate for much larger embeddings?

---

> ### Author Response · Authors · 2025-11-21
>
> > **Q1**: How resilient is the CREDIT verification scheme to this type of noise filtering attack?
>
> **R1**: Thank you for raising this important point. We agree that noise filtering attack could, in principle, reduce the variance of Gaussian noise. To evaluate this scenario, we conducted a dedicated experiment where we (i) fixed a total query budget $q$ that is sufficiently large for the adversary to train a functional surrogate model under a standard model extraction attack, and (ii) varied the repetition factor $m$, which reduces the number of distinct queries to $q/m$. The experimental results are shown below:
>
> | Repeat $m$ | Unique Query Ratio | Test Acc |
> |---------|---------------------|---------|
> |    1    |        0.1000       | 0.7303  |
> |    2    |        0.0500       | 0.5125  |
> |    4    |        0.0250       | 0.2833  |
> |    8    |        0.0125       | 0.1997  |
> |   10    |        0.0100       | 0.1720  |
>
> From the table, we observe a clear trend: increasing m dramatically degrades the attacker’s utility, because the number of unique queries collapses quickly. In other words, repeated queries consume the attacker’s query budget but do not provide additional information about the target model. As a consequence, averaging repeated queries becomes inherently expensive, and the resulting surrogate model becomes significantly less accurate. This makes such an attack economically impractical for the adversary. We have included these new experiments in Section D.2 of the revised version, and all modifications have been highlighted in blue.
>
> ---
>
> > **Q2**: Have the authors considered an adaptive attack where the adversary approximates the threshold and tunes their surrogate to bypass this estimated boundary?
>
> **R2**: Thank you for raising this scenario. The CREDIT threshold $\tau$ is not observable to the adversary. It is determined by defender specific parameters $(\sigma, \mathcal{Q}, |\mathcal{V}|, d)$ together with the desired rigor level, and none of these quantities are revealed through the query interface. As long as the model owner does not disclose these internal parameters or the verification outcome, the adversary has no information from which to approximate $\tau$. Therefore, it is not feasible for the adversary to tune a surrogate model to approach or bypass this hidden threshold.
>
> ---
>
> > **Q3**: Have the authors evaluated the computational overhead and reliability of the mutual information estimate for much larger embeddings?
>
> **R3**: Thank you for the question. For $n$ samples with $d$ dimensional embeddings, the KSG estimator's[1] complexity **scales linearly** in the embedding dimension. The overall computational overhead is on the order of
> $$O(n^{2} d),$$
> since the dominant operation is the kNN search in $d$ dimensional space. This makes the estimator practical even for fairly large embedding sizes. In our setting, $d = 1024$, which is already a relatively high dimensionality in both CV and NLP applications. For reference, widely used architectures adopt comparable or smaller embedding sizes (e.g., ViT-Base uses 768, and Qwen3-4B uses ~2048 for its hidden representation).
>
> ---
>
> We sincerely thank the reviewer for the time and effort. Your comments have been extremely valuable in improving the clarity and quality of the paper. If any further questions arise, please feel free to let us know, and we are eager to follow up promptly on all points.
>
> ----
> References
> [1]Kraskov, Alexander, Harald Stögbauer, and Peter Grassberger. "Estimating mutual information." Physical Review E—Statistical, Nonlinear, and Soft Matter Physics 69.6 (2004): 066138.

---

> > ### Author Response · Authors · 2025-11-26
> >
> > We sincerely thank the reviewer for the constructive feedback throughout the rebuttal process. We are confident that we have addressed all concerns raised in the initial review. In light of this, we respectfully hope that the reviewer may consider **raising the overall score**. We would be very glad to further engage in any additional discussion if the reviewer has remaining questions or concerns.

---

### Official Review · Reviewer_mf7r · 2025-10-30

**Soundness:** 4
**Presentation:** 3
**Contribution:** 4
**Rating:** 8
**Confidence:** 3

**Summary:**

This paper presents CREDIT, a certified defense framework against model extraction attacks (MEAs). It leverages mutual information (MI) to quantify similarity between neural network models and introduces a theoretically grounded threshold (CREDIT threshold) to verify ownership. The approach provides rigorous guarantees on false positive and false negative rates, and is empirically validated across image and graph modalities, consistently outperforming baselines.

**Strengths:**

1.	The paper provides a theoretical guarantee of the ownership verification, which is sound and novel.
2.	Mutual information is introduced to define the similarity between models.
3.	The proposed method achieves superior performance among baseline methods.

**Weaknesses:**

1.	The threat model is not clearly defined. What information can the defender have access to? The proposed method assumes access to the embedding representations of the suspicious model to estimate mutual information. This implies a white-box or at least gray-box setting. It is unclear whether CREDIT can operate effectively in a pure black-box scenario (e.g., only labels/soft labels are available).
2.	Definition 2 only shows how the functionality is replicated in a model. It does not directly answer the key question: “to what extent, does this similarity certify that h_sur has been extracted from the protected model?” This definition needs further clarification.
3.	The proposed framework relies on mutual information-based similarity measurement. However, mutual information may not fully capture the neural network behaviors. It would be great if the paper could discuss the scope and limitations of mutual information as a similarity measure in the defense.

**Questions:**

1.	What level of access does the defender require to the suspicious model?
2.	How does the CREDIT threshold quantitatively certify that a surrogate model has been extracted from the protected model, rather than merely being functionally similar?
3.	What are the limitations of using mutual information as a similarity metric between models?

---

> ### Author Response · Authors · 2025-11-21
>
> > **Q1**: What level of access does the defender require to the suspicious model?
>
> **R1**: Thank you for the question. The defender only requires black-box access to the suspicious model, where the defender can send queries and obtain the model’s output embeddings.
>
> ---
>
> > **Q2**: How does the CREDIT threshold quantitatively certify that a surrogate model has been extracted from the protected model, rather than merely being functionally similar?
>
> **R2**: Thank you for raising this point. CREDIT uses mutual information as a quantitative measure of how much information the suspicious model inherits from the protected model. The threshold $\tau(\sigma, \mathcal{Q})$, defined in Definition 2, separates models whose embeddings contain information exceeding what an independently trained model could possess (bounded by $\beta$ in Theorem 1).
>
> Therefore, if the estimated MI exceeds $\tau$, the method certifies that the suspicious model must have been extracted from the protected model rather than merely exhibiting functional similarity.
>
> ---
>
> > **Q3**: What are the limitations of using mutual information as a similarity metric between models?
>
> **R3**: Thank you for the question. A key limitation is that embeddings produced by deep neural networks do not follow a known or tractable distribution, so the mutual information cannot be computed in closed form. We must rely on an estimator, which inevitably introduces approximation error. In our work we adopt the widely used KSG estimator[1] for MI estimation in high dimensional embeddings.
>
> ---
>
> We sincerely thank the reviewer for the time and effort. Your comments have been extremely valuable in improving the clarity and quality of the paper. If any further questions arise, please feel free to let us know, and we are eager to follow up promptly on all points.
>
> ----
> References
> [1]Kraskov, Alexander, Harald Stögbauer, and Peter Grassberger. "Estimating mutual information." Physical Review E—Statistical, Nonlinear, and Soft Matter Physics 69.6 (2004): 066138.

---

> > ### Comment · Reviewer_mf7r · 2025-11-25
> >
> > Thank you for your response. My concerns have been well addressed. I will keep my positive score.

---

> > > ### Author Response · Authors · 2025-11-26
> > >
> > > We sincerely thank the reviewer for the positive assessment. We have additionally incorporated more comprehensive experiments and analyses in the revised version. We hope these updates further enhance your confidence in our contributions, the paper’s quality, and its impact.

---

### Official Review · Reviewer_ZRkj · 2025-10-31

**Soundness:** 3
**Presentation:** 2
**Contribution:** 2
**Rating:** 6
**Confidence:** 3

**Summary:**

This paper proposes a certified defense against MEAs, which formally formulates the problem of certified defense for MEAs and introduces the CREDIT method. The proposal aims to enable practical ownership verification while providing rigorous theoretical guarantees.

**Strengths:**

1. The first one that use the mutual information to quantify the dependency between two models.
2. The paper evaluates its method across two different data modalities and deploys CREDIT on four distinct backbone models within each modality to implement the defense methods.
3. A theoretical framework and sufficient experiments are proposed to support its conclusion.

**Weaknesses:**

1. The authors employ the Knowledge Distillation method (Romero et al., 2014) as the primary attack strategy. While this is a foundational technique, it is now considered relatively outdated. It would strengthen the paper's contribution to evaluate CREDIT against more recent and advanced model extraction attacks. A critical question remains: if challenged by these newer methods, can CREDIT still achieve reliable ownership verification while maintaining its rigorous theoretical guarantees?
2. In the model utility evaluation experiment, the authors did not provide detailed information on the number of suspicious models used when distinguishing between independent models and target models based on the same backbone models.

**Questions:**

1.What is the connection between mutual information and KSG?
2. How to calculate τ(σ, Q)?
3.Why does the threshold within the range [0, β] effectively separate the surrogate model hsur from independently trained models hind?

---

> ### Author Response · Authors · 2025-11-21
>
> > **Q1**: What is the connection between mutual information and KSG?
>
> **R1**: Thank you for the question. The embedding distributions produced by deep neural networks are unknown and do not admit closed-form densities. Therefore, we use the KSG estimator[1] as a standard nonparametric method for estimating mutual information directly from samples. KSG provides an accurate nearest-neighbor–based estimate of MI without requiring explicit knowledge of the underlying distribution. Thus, KSG is simply the estimator we use to compute the MI quantity defined in our method.
>
> ---
>
> > **Q2**: How to calculate τ(σ, Q)?
>
> **R2**: Thank you for the point. As described in Definition 2, the threshold
>
> $$
> \tau(\sigma, Q) = \beta \left[ 1 - \rho \exp\left(- \frac{Q\beta}{\eta\, d \,|\mathcal{V}|}\right) \right]
> $$
>
> is obtained directly by plugging the mutual-information upper bound $\beta$ from Theorem 1 into the verification formulation. Given $\sigma$ (which determines $\beta$), the query budget Q, embedding dimension d, and the size of the verification set $|\mathcal{V}|$, $\tau$ is computed in closed form.
>
> ---
>
> > **Q3**: Why does the threshold within the range [0, $\beta$] effectively separate the surrogate model $h_{\text{sur}}$ from independently trained models $h_{\text{ind}}$?
>
> **R3**: Thank you for the question. Theorem 1 states that $\beta$ is an upper bound on the mutual information between any model and the defended model. Surrogate models $h_{\text{sur}}$, which directly replicate the behavior of the defended model, achieve MI values close to this upper bound. In contrast, independently trained models $h_{\text{ind}}$ exhibit much smaller MI values.
>
> Once $\tau$ is computed (Definition 2), setting the separation threshold within [0, $\beta$] reliably distinguishes the high-MI surrogate models from the low-MI independent models. Thus, $\tau$ provides an effective and theoretically grounded separator between $h_{\text{sur}}$ and $h_{\text{ind}}$.
>
> ---
>
> We sincerely thank the reviewer for the time and effort. Your comments have been extremely valuable in improving the clarity and quality of the paper. If any further questions arise, please feel free to let us know, and we are happy to follow up promptly on all points.
>
> ----
> References
> [1]Kraskov, Alexander, Harald Stögbauer, and Peter Grassberger. "Estimating mutual information." Physical Review E—Statistical, Nonlinear, and Soft Matter Physics 69.6 (2004): 066138.

---

> > ### Comment · Reviewer_ZRkj · 2025-11-25
> >
> > Thank you for the clarification. I will keep my current positive score.

---

> > > ### Author Response · Authors · 2025-11-26
> > >
> > > We sincerely thank the reviewer for the positive assessment. We have additionally incorporated more comprehensive experiments and analyses in the revised version. We hope these updates further enhance your confidence in our contributions, the paper’s quality, and its impact. In light of this, we respectfully hope you may consider **raising the overall rating**. And we would be happy to clarify any further points if needed.

---

### Official Review · Reviewer_d192 · 2025-11-05

**Soundness:** 3
**Presentation:** 2
**Contribution:** 2
**Rating:** 4
**Confidence:** 4

**Summary:**

This paper introduces CREDIT (Certified Defense of Deep Neural Networks against Model Extraction Attacks), a certified defense framework designed to protect models from Model Extraction Attacks (MEAs) in the Machine Learning as a Service (MLaaS) setting. The work adds Gaussian noise to model embeddings, which enables the derivation of a mutual information (MI) upper bound between the protected model and any surrogate model. Based on this bound, the authors define a CREDIT threshold to verify model ownership with formal probabilistic guarantees. Finally, theoretical analysis provides provable bounds on Type I and Type II errors that decay exponentially with the verification set size.

**Strengths:**

1 Quantifying model similarity is crucial for model protection, and this work is built on solid, established theory by employing Mutual Information as an information-theoretic metric for model similarity and the Gaussian Mechanism from differential privacy for provable bounds.

2 Unlike traditional approaches based on model watermarking or fingerprinting, this work has minimal impact on the model's performance.

3 Furthermore, this work rigorously derives the theoretical upper bound of mutual information, which substantiates the reliability and credibility of the proposed work.

**Weaknesses:**

First, in the MLaaS setting, services only expose the final outputs, and introducing noise perturbations into intermediate layer embeddings represents a non-standard and highly demanding assumption.

Second, the experimental validation is limited, lacking comparisons with other metrics which can quantify model similarity, and performance on large language models (LLMs).

Finally, mutual information estimation is critical for this work’s effectiveness, but the KSG estimator is known to be unstable and computationally intensive in high-dimensional spaces.

**Questions:**

1.Choice of σ：Does the σ need to be selected differently for different task models or tasks? If so, would this lead to a high search cost? When considering efficiency in experiments, is this time cost taken into account?

2.How robust is CREDIT to query-averaging/denoising attacks? If an adversary issues m repeated queries per input and averages responses to reduce Gaussian noise variance, can they increase MI and evade detection? What defenses (rate-limit, per-user noise, query budget accounting) do you recommend?

3.KSG estimator hyperparameter: How is the number of nearest neighbors set in the KSG estimator, and how significantly does this parameter affect mutual information computation?

4.Lemma 5 upper bound: How is the upper bound in Lemma 5 derived?

---

> ### Author Response · Authors · 2025-11-21
>
> > **Q1**: Choice of $\sigma$：Does the $\sigma$ need to be selected differently for different task models or tasks? If so, would this lead to a high search cost? When considering efficiency in experiments, is this time cost taken into account?
>
>
> **R1**: Thank you for the question. $\sigma$ is indeed a model-specific parameter. As discussed in Section 3.3 there is no closed-form expression, we evaluate a set of candidate $\sigma$ values and directly compute our objective to obtain the optimal $\sigma^\star$.
>
> As reported in our experiments in Section 4.3, as shown in Table 3, the grid-search time for selecting $\sigma$ is included in the preparation-time measurement. Besides, model inference time and comparing time are included in the verification-time. Overall, $\sigma$ selection is efficient, stable across different task models, and exhibits consistent behavior in all our experiments.
>
> ---
> > **Q2**: How robust is CREDIT to query-averaging/denoising attacks? If an adversary issues $m$ repeated queries per input and averages responses to reduce Gaussian noise variance, can they increase MI and evade detection? What defenses (rate-limit, per-user noise, query budget accounting) do you recommend?
>
> **R2**: Thank you for raising this important point. We agree that repeated queries and response averaging could, in principle, reduce the variance of Gaussian noise. To evaluate this scenario, we conducted a dedicated experiment where we (i) fixed a total query budget $q$ that is sufficiently large for the adversary to train a functional surrogate model under a standard model extraction attack, and (ii) varied the repetition factor $m$, which reduces the number of distinct queries to $q/m$. The experimental results are shown below:
>
> | Repeat $m$ | Unique Query Ratio | Test Acc |
> |-----|--------|-------|
> |    1    |    0.1000  | 0.7303  |
> |    2    |    0.0500  | 0.5125  |
> |    4    |    0.0250  | 0.2833  |
> |    8    |    0.0125  | 0.1997  |
> |   10    |   0.0100  | 0.1720  |
>
> From the table, we observe a clear trend: increasing m dramatically degrades the attacker’s utility, because the number of unique queries collapses quickly. In other words, repeated queries consume the attacker’s query budget but do not provide additional information about the target model. As a consequence, averaging repeated queries becomes inherently expensive, and the resulting surrogate model becomes significantly less accurate. This makes such an attack economically impractical for the adversary. We have included these new experiments in Section D.2 of the revised version, and all modifications have been highlighted in blue.
>
> ---
>
> > **Q3**: KSG estimator hyperparameter: How is the number of nearest neighbors set in the KSG estimator, and how significantly does this parameter affect mutual information computation?
>
> **R3**: Thank you for the question. The number of nearest neighbors is indeed an important hyper parameter in the KSG estimator. To evaluate its impact, we conducted an additional set of experiments on CIFAR-100, where we varied the value of $k \in ${3, 5, 7, 10} while keeping all other settings unchanged. Our main experiments use $k = 5$. The results are shown below:
>
> | k  | MI (mean ± std) |
> |----|-----|
> | 3  | 0.0492 ± 0.0028  |
> | 5  | 0.0422 ± 0.0021  |
> | 7  | 0.0402 ± 0.0026   |
> | 10 | 0.0470 ± 0.0025   |
>
> The results indicate that, within a reasonable range of $k$, the estimated mutual information remains highly stable. Our choice of $k = 5$ reflects a balance between efficiency, since a small number of neighbors reduces computation cost, and robustness, since it avoids relying on very unique neighbors that may introduce instability in the estimate. We have included these additional experiments in Appendix D.1 and marked in blue, where further details are provided.
>
> ---
>
> > **Q4**: Lemma 5 upper bound: How is the upper bound in Lemma 5 derived?
>
> **R4**: Thank you for the question. The upper bound in Lemma 5 follows directly from Theorem 4 and Definition 4. First, by Theorem 4 we know that for the input X and its Gaussian mechanism output $Z = e_g(X)$, the mutual information is bounded as
>
> $$I(X, Z) \le \beta.$$
>
> Next, Lemma 5 states that $Y = e_h(X)$ and $Z = e_g(X)$ are conditionally independent given $X$. This implies the Markov chain
> $$Y \rightarrow X \rightarrow Z.$$
>
> By the Definition 4, this gives
> $$I(Y, Z) \le I(X, Z).$$
>
> Combining the two results, we obtain
> $$
> I(e_h(X), e_g(X)) = I(Y, Z) \le I(X, Z) \le \beta,
> $$
> which is exactly the claimed upper bound in Collary 6.
>
> ---
>
> We sincerely thank the reviewer for the time and effort. Your comments have been extremely valuable in improving the clarity and quality of the paper. If any further questions arise, please feel free to let us know, and we are happy to follow up promptly on all points.

---

> > ### Author Response · Authors · 2025-11-26
> >
> > > **W1**: First, in the MLaaS setting, services only expose the final outputs, and introducing noise perturbations into intermediate layer embeddings represents a non-standard and highly demanding assumption.
> >
> > **To W1**:
> > We appreciate the reviewer’s comment. In fact, using embedding representations is highly standard and widely adopted in modern MLaaS ecosystems. Embeddings capture the intrinsic geometric and semantic structure learned by the model, and many contemporary foundation models—including GPT-style embedding models [1]—are explicitly designed so that their intermediate representations function as semantically meaningful vectors. These embeddings are routinely exposed or utilized in downstream services such as information retrieval [2], semantic search, similarity matching, and recommendation systems [3].
> >
> > Therefore, incorporating noise at the embedding level does not impose an unrealistic assumption. Rather, it aligns with how real-world MLaaS pipelines leverage intermediate representations as primary service outputs, and our method naturally operates within this widely prevalent model-embedding paradigm.
> >
> >
> > ----
> >
> > > **W2**: Second, the experimental validation is limited, lacking comparisons with other metrics which can quantify model similarity, and performance on large language models (LLMs).
> >
> > **To W2**: Thank you very much for this question. Regarding metrics that quantify model similarity, we would like to reiterate that the contribution of our work lies in performing certified ownership verification based on mutual information (MI). Owing to the Gaussian mechanism used in our method, we obtain a theoretical upper bound on MI, which then serves as a certificate for verifying ownership of suspicious models.
> >
> > Concerning comparisons with LLMs, we sincerely appreciate the reviewer for raising this point. To keep the evaluation lightweight and controlled, we additionally trained a Word2Vec[4] model on the STSb dataset[5] and applied our CREDIT framework to verify ownership of suspicious models. The results are shown below:
> >
> > | Model     |    MI     |  AUC   |
> > |-----------|-----------|--------|
> > | ResNet    | 2.7470    | 1.0000 |
> > | GCN       | 1.6743    | 1.0000 |
> > | Word2Vec  | 1.2331    | 1.0000 |
> >
> > From the table, we observe that CREDIT continues to achieve consistent and effective ownership verification, which further demonstrates that our method is broadly applicable not only to CV and graph modalities but also to NLP modalities. We have included these new experiments in Section D.5 of the revised version, and all modifications have been highlighted in blue.
> >
> >
> > ----
> >
> > > **W3**: Finally, mutual information estimation is critical for this work’s effectiveness, but the KSG estimator is known to be unstable and computationally intensive in high-dimensional spaces.
> >
> > **To W3**: Thank you for the question. For $n$ samples with $d$ dimensional embeddings, the KSG estimator's[6] complexity **scales linearly** in the embedding dimension. The overall computational overhead is on the order of $O(n^{2} d),$ since the dominant operation is the kNN search in $d$ dimensional space. This makes the estimator practical even for fairly large embedding sizes.
> >
> > In our setting, $d = 1024$, which is already a relatively high dimensionality in both CV and NLP applications. For reference, widely used architectures adopt comparable or smaller embedding sizes (e.g., ViT-Base uses 768, and Qwen3-4B uses ~2048 for its hidden representation).
> >
> > ----
> > References
> > [1]Tao, Chongyang, et al. "Llms are also effective embedding models: An in-depth overview." arXiv preprint arXiv:2412.12591 (2024).
> > [2]Palangi, Hamid, et al. "Deep sentence embedding using long short-term memory networks: Analysis and application to information retrieval." IEEE/ACM transactions on audio, speech, and language processing 24.4 (2016): 694-707.
> > [3]Zhang, Fuzheng, et al. "Collaborative knowledge base embedding for recommender systems." Proceedings of the 22nd ACM SIGKDD international conference on knowledge discovery and data mining. 2016.
> > [4]Mikolov, Tomas, et al. "Efficient estimation of word representations in vector space." arXiv preprint arXiv:1301.3781 (2013).
> > [5]Cer, Daniel, et al. "Semeval-2017 task 1: Semantic textual similarity-multilingual and cross-lingual focused evaluation." arXiv preprint arXiv:1708.00055 (2017).
> > [6]Kraskov, Alexander, Harald Stögbauer, and Peter Grassberger. "Estimating mutual information." Physical Review E—Statistical, Nonlinear, and Soft Matter Physics 69.6 (2004): 066138.

---

> > > ### Author Response · Authors · 2025-11-26
> > >
> > > We sincerely thank the reviewer for the constructive feedback throughout the rebuttal process. We are confident that we have addressed all concerns raised in the initial review. In light of this, we respectfully hope that the reviewer may consider **raising the overall score**. We would be very glad to further engage in any additional discussion if the reviewer has remaining questions or concerns.

---

### Author Response · Authors · 2025-12-01

Dear AC,

We sincerely thank the AC for the time and effort devoted to handling our submission, especially given the heavy workload during the rebuttal period. Below we summarize the key weaknesses raised by the reviewers and how our revisions directly address them.


**Strengths**
1. The work establishes CREDIT as the first framework that uses mutual information to quantify dependency between models and provide certified ownership verification guarantees. (Reviewer d192, ZRkj, mf7r, JkN2, fnV4)
2. The paper offers rigorous theoretical analysis, including a provable mutual information upper bound, tightness proofs, and strict probabilistic guarantees on Type I and Type II errors. (Reviewer d192, mf7r, fnV4)
3. CREDIT imposes minimal impact on model accuracy and does not require training any additional models, making it practically efficient and lightweight. (Reviewer d192, JkN2)
4. The experimental evaluation is extensive across multiple modalities and diverse backbone architectures, consistently demonstrating strong generalization and superior empirical performance. (Reviewer ZRkj, JkN2, fnV4, mf7r)


**Weaknesses**
1. Whether repeated queries and output averaging could weaken CREDIT’s effectiveness. (Reviewer d192, JkN2)
2. Whether the KSG estimator is sensitive to the choice of neighborhood size and computationally expensive at higher embedding dimensions. (Reviewer d192, JkN2)
3. Whether CREDIT remains effective under worst-case adversarial manipulations that reduce utility. (Reviewer fnV4)
4. Whether the term “defense” accurately reflects our contribution compared to the more precise framing as ownership verification. (Reviewer fnV4)


**Our revisions to address the weaknesses**

**1. Query-averaging and denoising attacks:** We performed new experiments that fix the total query budget and vary the repetition factor. Increasing repetition sharply reduces the number of distinct queries, making the extracted surrogate model lose utility and become unusable. These results demonstrate that averaging-based denoising is economically and practically infeasible. These experiments have been added to Appendix D.2 and highlighted in blue.

**2. KSG estimator sensitivity and computational cost:** We conducted additional analyses showing that the estimator is stable across different neighborhood sizes. We further provided a complexity analysis proving that KSG scales linearly with embedding dimension. Since our experiments already use high-dimensional embeddings, CREDIT remains substantially more efficient than all baselines. These updates appear in Appendix D.1 and are marked in blue.

**3. Worst-case certified guarantees:** We constructed experiments where the adversary can reduce utility within an allowed range while aggressively trying to decorrelate surrogate embeddings. Even under such worst-case manipulations, CREDIT maintains strong verification performance. These results are included in Appendix D.4 and highlighted in blue.

**4. Terminology adjustment:** Following the suggestion of Reviewer `fnV4`, we revised the manuscript to frame CREDIT as an ownership verification method rather than a defense, which more precisely reflects our technical contribution.

Reviewers have confirmed that our responses have resolved the raised concerns. We believe these revisions fully address all reviewer comments and substantially enhance the contribution and overall quality of the paper. We sincerely thank the AC for the time and effort devoted to handling our submission.


Best regards,
The Authors

---

### Meta-Review · Area_Chair_Ud9p · 2026-01-07

**Summary:**

This paper presents a method for certified ownership verification. It adds Gaussian noise to the embeddings of the model. If the mutual information between the embeddings of suspect model and defended model is higher than a certain threshold, then it is determined as a Model Extraction Attack. The author demonstrated the effectiveness of the proposed method from both theoretical and experimental perspectives, and also showed that it can maintain the utility of the model.

The reviewers all acknowledged the theoretical contributions of this paper, but also raised several concerns. The main concerns are:
1) The KSG estimator requires a large number of samples in high-dimensional space to accurately estimate mutual information. This involves a significant computational cost and is unreliable.
2) An adaptive attacker, i.e., an attacker who knows the author's defense strategy, may bypass the defense. For instance, it can repeatedly query to get the average output.
3) The authors claim that they have proposed a defense against model extraction attacks, but in practice, they cannot prevent any attacks. They can only detect whether an attack has occurred.

The author actively addressed the concerns raised by the reviewers. They conducted new experiments to respond to (1), demonstrating that the KSG estimator remains efficient and reliable in high-dimensional spaces. In response to (2), they demonstrated through experiments that averaging the repeated queries is ineffective. They revised certain expressions, changing "defense" to "certified ownership verification" in response to (3).

However, Concern (3) is still outstanding. Although the author replaced the term "defense" with "certified ownership verification", this does not mean that the concern has been solved. The change in core concepts implies that the related work, experimental settings, evaluation metrics, and comparison baselines of this paper should all be modified accordingly. The worst-case robustness and unremovability, which are the certified ownership verification focuses on, should also be included in the research.

In addition, the experimental setup is weak and outdated. The datasets CIFAR-10 and CIFAR-100 are too simple and low-dimension. The baselines in image classification were all published before 2022. Backbone does not include the currently most popular Transformer architecture. I suggest intensifying the experiments.

Therefore, this article is not yet suitable for publication at present. I hope that the reviewers' comments will help the authors to further revise this paper.

**Reviewer Concerns:**

Concern (3) is still outstanding. Although the author replaced the term "defense" with "certified ownership verification", this does not mean that the concern has been solved. The change in core concepts implies that the related work, experimental settings, evaluation metrics, and comparison baselines of this paper should all be modified accordingly. The worst-case robustness and unremovability, which are the certified ownership verification focuses on, should also be included in the research.

**Reviewer Scores:**

Reviewer d192 may increase score because his concerns have been addressed.

Reviewers ZRkj and mf7r will maintain their initial scores.

Reviewers JkN2 and fnV4 will not increase scores because their concerns have not been fully addressed.

---

### Decision · Program_Chairs · 2026-01-26

Reject